# OMP: One-step Meanflow Policy with Directional Alignment

**Han Fang** [1]  **Yize Huang** [2]  **Yuheng Zhao** [1]  **Paul Weng** [3]  **Xiao Li** [2]  **Yutong Ban** [1]

## Abstract

Robot manipulation has increasingly adopted data-driven generative policy frameworks, yet the field faces a persistent trade-off: diffusion models suffer from high inference latency, while flow-based methods often require complex architectural constraints. Although in image generation domain, the MeanFlow paradigm offers a path to single-step inference, its direct application to robotics is impeded by critical theoretical pathologies, specifically spectral bias and gradient starvation in low-velocity regimes. To overcome these limitations, we propose the One-step MeanFlow Policy (OMP), a novel framework designed for high-fidelity, real-time manipulation. We introduce a lightweight directional alignment mechanism to explicitly synchronize predicted velocities with true mean velocities. Furthermore, we implement a Differential Derivation Equation (DDE) to approximate the Jacobian-Vector Product (JVP) operator, which decouples forward and backward passes to significantly reduce memory complexity. Extensive experiments on the Adroit and Meta-World benchmarks demonstrate that OMP outperforms state-of-the-art methods in success rate and trajectory accuracy, particularly in high-precision tasks, while retaining the efficiency of single-step generation.

## 1. Introduction

Robot manipulation is a cornerstone of embodied AI, enabling diverse applications from household chores to industrial assembly (Rajeswaran et al., 2018; Yu et al., 2019; Zhao et al., 2023). The field has recently shifted from supervised regression to data-driven generative frameworks that model complex, multimodal distributions (Chi et al., 2023; Zhang

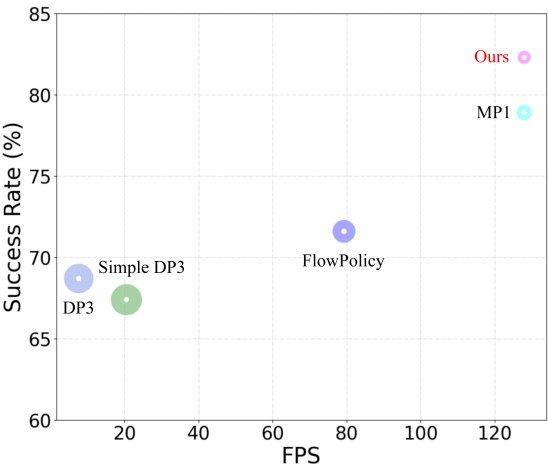

*Figure 1.* **Inference Speed vs. Success Rate.** Comparison of OMP against SOTA diffusion and flow baselines on Adroit and Meta-World tasks. The x-axis denotes control frequency (FPS), the y-axis shows average success rate, and circle radii represent standard deviation.

et al., 2024). Within the broader landscape of generative robot learning, Diffusion models, such as Diffusion Policy (DP) (Chi et al., 2023) and its 3D extension DP3 (Ze et al., 2024), have led this advance by formulating control as a probabilistic denoising process. Recent innovations have further enhanced this approach by incorporating geometric symmetries (Yang et al., 2024a) or scaling via bimanual manipulation transformers (Liu et al., 2024b). However, despite their high-fidelity generation, these methods rely on iterative denoising with high function evaluation counts, resulting in latency that limits high-frequency control.

To overcome this latency bottleneck, Flow-based methods and Consistency models have emerged, employing techniques like AdaFlow (Hu et al., 2024), FlowPolicy (Zhang et al., 2024), and ManiFlow (Yan et al., 2025). These approaches leverage Ordinary Differential Equations (ODEs) or consistency distillation (Song et al., 2023) to transform noise into actions via learned vector fields, enabling rapid single-step inference. While they effectively mitigate delays, they often require complex training pipelines involving segmented flows or explicit constraints. These architectural demands can over-constrain the model, creating a persistent trade-off between the ease of deployment and the ability to generalize to novel scenes (Sheng et al., 2025).

[1]Global College, Shanghai Jiao Tong University, Shanghai, China [2]School of Mechanical Engineering, Shanghai Jiao Tong University, Shanghai, China [3]Duke Kunshan University, Jiangsu, China. Correspondence to: Yutong Ban <yban@sjtu.edu.cn>.

*Proceedings of the 43rd International Conference on Machine Learning*, Seoul, South Korea. PMLR 306, 2026. Copyright 2026 by the author(s).

A recent theoretical breakthrough, MeanFlow (Geng et al., 2025), offers a potential unification by circumventing ODE solvers entirely, instead modeling interval-averaged velocity through the MeanFlow Identity. Its application to robotics, MP1 (Sheng et al., 2025), achieves single-step inference (NFE = 1) with a latency of just 6.8 ms while surpassing FlowPolicy in success rates. However, our rigorous analysis reveals that a direct application of MeanFlow to robotic manipulation is impeded by three critical theoretical pathologies. First, we identify *Spectral Bias*, where the MeanFlow objective's integral formulation functions as a low-pass filter that facilitates coarse guidance but attenuates the rapid adjustments necessary for precision. Second, in high-precision tasks requiring slow, fine-grained movements, the standard objective suffers from *Gradient Starvation*, where the optimization landscape provides negligible directional guidance in low-velocity regimes. Finally, the reliance on the exact Jacobian-Vector Product (JVP) operator necessitates nested derivative computations, leading to a *Memory Complexity* that scales with tangent dimensionality, thereby prohibiting the training of large-scale backbones on standard hardware.

To resolve these limitations, we propose OMP, a novel framework designed for efficient, high-fidelity manipulation without the architectural overhead of prior methods. By introducing a directional alignment to decouple directional learning from magnitude, and by optimizing the computational graph via finite-difference approximations, we enable robust single-step generation that retains the precision of diffusion models with the speed of flow-based methods.

Our contributions are summarized as follows:

- To enable the application of the Meanflow paradigm to robot policy learning, we identify critical bottlenecks, specifically the prevalence of spectral bias and gradient starvation in low-velocity regimes, compounded by prohibitive memory overheads in standard Jacobian computations that fundamentally restrict the scalability of complex networks.

- We introduce a novel framework to transcend these limitations, featuring a Directional Alignment mechanism that ensures robust convergence by explicitly synchronizing predicted interval-averaged velocities with true mean velocity to overcome spectral bias and gradient starvation, alongside a novel Differential Derivation Equation (DDE) that achieves high memory efficiency via finite difference approximation.

- We conduct extensive evaluations on the Adroit and Meta-World benchmarks, as well as real-world robotic tasks. The empirical results demonstrate that OMP outperforms state-of-the-art methods, including MP1 and FlowPolicy, establishing a new standard for real-time generative robot control.

## 2. Related Work

In this section, we review the literature most pertinent to our study, focusing on generative policy learning and visual representations in robotics. For a more comprehensive discussion, please refer to Section A.

### 2.1. Generative Models for Robot Policy Learning

Generative modeling has transformed robot learning by treating action generation as a probabilistic process. This shift runs parallel to foundational advances in media generation (Polyak et al., 2025) and scaling rectified flows (Esser et al., 2024). The current landscape is largely defined by the trade-off between the high-fidelity generation of Diffusion models and the inference efficiency of Flow-based methods.

**Diffusion Models.** Diffusion Policy (DP) (Chi et al., 2023) and its 3D extension, DP3 (Ze et al., 2024), treat policy learning as a conditional denoising process, achieving high success rates by iteratively refining action trajectories. Recent works have refined this paradigm: HDP (Ma et al., 2024) and RDT (Liu et al., 2024b) enhance spatial modeling and dual-arm manipulation, respectively, while Equivariant Diffusion Policy (Wang et al., 2024a) and EquiBot (Yang et al., 2024a) incorporate geometric symmetries to improve sample efficiency. Others focus on architectural optimizations, such as representation alignment in REPA (Yu et al., 2025) and transformer modulation in MTDP (Wang et al., 2025a). Despite their performance, diffusion models inherently suffer from high inference latency due to the multi-step denoising requirement (e.g., DP3 often requires NFE=10), creating a bottleneck for real-time control.

**Flow-based and One-Step Models.** To address latency, flow-based models learn invertible mappings via vector fields, building on Flow Matching (Lipman et al., 2023) and Rectified Flow (Liu et al., 2022). Efforts like Consistency Policy (Prasad et al., 2024) and OneDP (Wang et al., 2024c) distill multi-step diffusion policies into one-step solvers. More recently, Consistency Flow Matching (Yang et al., 2024b) has enabled straight-line flow generation, a technique adopted by ManiCM (Lu et al., 2024) and ManiFlow (Yan et al., 2025). FlowPolicy (Zhang et al., 2024) integrates 3D point clouds with consistency flow matching to achieve single-step inference. However, these methods often require auxiliary consistency constraints during training to ensure trajectory validity.

The most relevant predecessors to our work are MeanFlow-based paradigms, which offer a clearer path to efficiency by averaging velocity fields. MP1 (Sheng et al., 2025) adapts MeanFlow (Geng et al., 2025) to robotics, eliminating ODE solvers to achieve 6.8ms latency, while DM1 (Zou et al., 2025) utilizes dispersive regularization for one-step manip-

ulation. Critically, both MP1 and DM1 rely on objectives where the predicted velocity direction may drift from the true mean velocity. Our method, OMP, addresses this by introducing *Directional Alignment* to enforce vector consistency and utilizing a Differential Derivation Equation (DDE) to approximate the Jacobian-Vector Product (JVP). This allows for efficient finite-difference computation (Karakida et al., 2023), improving trajectory accuracy and reducing memory consumption compared to DM1 and MP1.

### 2.2. Visual Representations for Manipulation

Robot policy learning relies heavily on the choice of input modality to perceive environmental states. While traditional approaches often utilized explicit state estimation, modern data-driven frameworks have shifted toward raw visual observations. Methods such as BC-Z (Jang et al., 2022) utilize 2D RGB or depth images, aligning visual-language features for generalization. ALOHA (Zhao et al., 2023) employs Action Chunking with Transformers (ACT) to model relationships between 2D vision and action sequences, while HPT (Wang et al., 2024b) scales proprioceptive-visual learning using heterogeneous transformers. However, 2D inputs often lack the explicit spatial information required for high-precision manipulation tasks.

To resolve spatial ambiguity, 3D modalities have become prevalent. Earlier 3D approaches, such as PerACT (Shridhar et al., 2022) and ACT3D (Gervet et al., 2023), utilized voxel-based representations but incurred high memory costs. Consequently, recent methods like RVT (Goyal et al., 2023) and RVT-2 (Goyal et al., 2024) leverage Farthest Point Sampling (FPS) on point clouds to preserve spatial structure efficiently. This paradigm has been further enhanced by integrating language-aligned 3D keypoints, as seen in CLAP (Hu et al., 2026), or by structuring point cloud processing for robust multi-modal imitation, as in PointMapPolicy (Jia et al., 2025) and Match Policy (Huang et al., 2025). Beyond point clouds, Gaussian Splatting has recently been adopted for scalable world modeling (Lu et al., 2025a) and one-shot manipulation (Yang et al., 2025), offering a continuous 3D representation that balances fidelity and efficiency.

## 3. Background

**Diffusion Model.** Diffusion models (DM) are generative models that model $p_{\text{data}}(\mathbf{x})$ via stochastic forward/reverse diffusion. The forward process adds noise to $\mathbf{x}_1 \sim p_{\text{data}}(\mathbf{x})$ over $T$ steps to reach $\mathbf{x}_T \sim \mathcal{N}(\mathbf{0}, \mathbf{I})$:

$$\mathbf{x}_t = \sqrt{\alpha_t}\mathbf{x}_{t-1} + \sqrt{1 - \alpha_t}\boldsymbol{\epsilon}_t, \tag{1}$$

where $\alpha_t$ is a noise schedule and the noise $\boldsymbol{\epsilon}_t \sim \mathcal{N}(\mathbf{0}, \mathbf{I})$. The reverse process learns $\boldsymbol{\epsilon}_\theta(\mathbf{x}_t, t)$ to invert diffusion, minimizing $\|\boldsymbol{\epsilon} - \boldsymbol{\epsilon}_\theta(\mathbf{x}_t, t)\|^2$. DM models multimodal distributions but suffer from slow sampling.

Flow matching (FM) addresses DM inefficiency via a deterministic flow $\mathbf{f}(\mathbf{x}, t)$ transforming $p_0(\mathbf{x})$ to $p_{\text{data}}(\mathbf{x}) = p_1(\mathbf{x})$. The flow satisfies the continuity equation:

$$\frac{\partial p_t(\mathbf{x})}{\partial t} = -\nabla \cdot (p_t(\mathbf{x})\mathbf{f}(\mathbf{x}, t)), \tag{2}$$

where $p_t(\mathbf{x})$ is the distribution at time $t$. FM enables fast sampling, which usually contains tens of ODE steps.

These methods optimize DMs for sampling in $T = 1$ or $T \ll 100$ steps. Recently, one-step models have drawn increase attention for its faster sampling speed. Those one-step models use a decoder to map $\mathbf{x}_T$ directly to $\mathbf{x}_1$. They balance speed and quality, making them suitable for robotic tasks that require low latency.

**Diffusion-based Policy.** Diffusion-based policies adapt diffusion models to policy learning, modeling state-to-action distribution $\pi(\mathbf{a} \mid \mathbf{s})$ as the target of a state-conditioned diffusion process. Actions are sampled via ODE solving or direct decoding. These policies learn from demonstrations and can handle high-dimensional actions.

## 4. Method

Our work builds upon the MeanFlow framework and constructs a robot learning paradigm designed to achieve single-step trajectory generation directly from 3D point-cloud inputs. While the original MeanFlow implementation successfully eliminates ODE solver errors by learning interval-averaged velocities, we identify specific theoretical pathologies that hinder its application in high-precision robotic manipulation. In this section, we first formalize the MeanFlow paradigm. Subsequently, we analyze three critical limitations: spectral bias, gradient starvation in high-dimensional error landscapes, and the memory complexity of higher-order differentiation. Finally, we introduce the One-step MeanFlow Policy (OMP), which overcomes these barriers through Directional Alignment and a Differential Derivation Equation.

### 4.1. Preliminaries: The MeanFlow Framework

We define the necessary notation for the MeanFlow framework. The core innovation of MeanFlow is the direct estimation of interval-averaged velocities $u(z_t, r, t)$, departing from the instantaneous velocity modeling standard in flow matching. This formulation circumvents the computational burden of multi-step numerical integration, enabling single-step inference (NFE = 1) which is essential for real-time control frequencies.

The inference process recovers a clean state $z_0$ from an initial noise distribution $z_T \sim \mathcal{N}(0, I)$ in a single step. We define the true mean velocity $v_0$ as the vector connecting

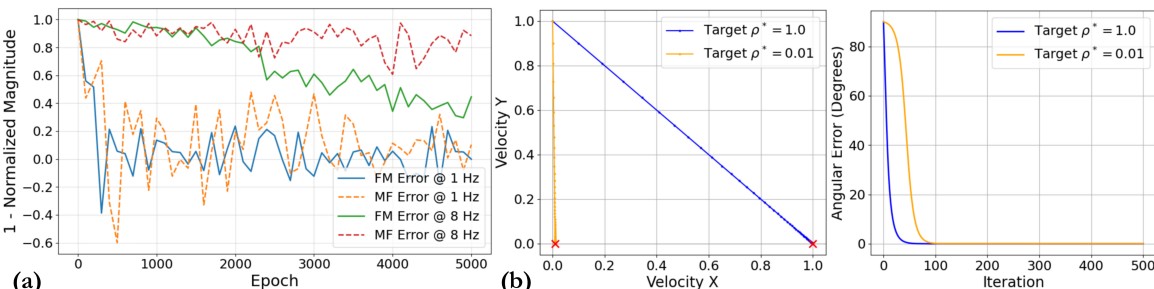

*Figure 2.* **(a)** Frequency-specific convergence rate for Flow Matching (FM) and MeanFlow (MF) models on a synthetic multi-tone velocity field. **(b)** The Optimization trajectories (left) and their corresponding angular error over training iterations (right) of a learned velocity vector in 2D space as it regresses to a high-magnitude target ($\rho^* = 1.0$, blue) versus a low-magnitude target ($\rho^* = 0.01$, orange). See more details about preliminary experimental setup in Section D.

the noise to the target state:

$$v_0 \triangleq z_T - z_0. \tag{3}$$

During training, the parameterized model $u_\theta(z_t, r, t|c)$, conditioned on context $c$, is optimized via the MeanFlow Identity to predict the average velocity between randomly sampled timesteps $t$ and $r$. This identity relates the total derivative of the velocity to the discrepancy between instantaneous and average velocities:

$$u(z_t, r, t|c) = v(z_t, t|c) - (t - r)\frac{d}{dt}u(z_t, r, t|c). \tag{4}$$

The right-hand side of Equation (4) constitutes the target velocity $u_{tgt}$. The standard objective minimizes the Mean Squared Error (MSE) between the prediction and this target. MP1 (Sheng et al., 2025) augments this loss by a Dispersive Loss $\mathcal{L}_{Disp}$ (Wang & He, 2025) to encourage latent space feature discrimination.

### 4.2. Theoretical Analysis of MeanFlow in Robotic Policies

We analyze why the standard MeanFlow formulation may fail when applied directly to some robotic manipulation scenarios. We identify three distinct barriers and analyze them in detail in Section B.

#### 4.2.1. SPECTRAL BIAS

The first challenge stems from the spectral bias inherent in the MeanFlow formulation. The average velocity is defined as the time-normalized integral of instantaneous dynamics over a sampled interval $[r, t]$:

$$u(t) = \frac{1}{t - r}\int_r^t v(\tau)d\tau. \tag{5}$$

Because this operation integrates strictly over time along the denoising trajectory, it exhibits a distinct frequency response. In the frequency domain, time integration corresponds to

division by $i\omega$. Under a localized wide-sense stationary (WSS) assumption, the Power Spectral Density (PSD) of the target signal $S_u(\omega)$ decays quadratically relative to the source dynamics $S_v(\omega)$:

$$S_u(\omega) \propto \frac{S_v(\omega)}{\omega^2}. \tag{6}$$

This $1/\omega^2$ decay demonstrates that the temporal averaging operator acts as a low-pass filter, aggressively attenuating high-frequency directional adjustments along the trajectory.

To empirically validate this temporal filtering effect, we model the denoising trajectory as a synthetic sine wave and compare the spectral error of standard flow matching against MeanFlow models (Figure 2a). Standard flow matching accurately captures high-frequency (8 Hz) trajectories. In contrast, MeanFlow exhibits persistently high error on the same 8 Hz paths. The MeanFlow objective attenuates the 8 Hz signal by a factor of approximately $1/64$ relative to a 1 Hz baseline, effectively reducing the high-frequency target to noise. At the lower 1 Hz frequency, both models converge, though MeanFlow displays higher variance. Additional preliminary results could be checked in Section C.2.1.

This analysis highlights a fundamental limitation: compressing multi-step denoising into a single step provides only coarse, low-frequency guidance. The straight-line approximation discards the high-frequency non-linearities necessary to accurately navigate curved vector fields. Therefore, while a filtered target yields a general approximation, minimizing the generation error:

$$\mathcal{L}_{MSE} = \|u_\theta(z_t, r, t|c) - u_{\text{tgt}}\| \tag{7}$$

dictates that the learned policy must overcome this temporal spectral bias. Achieving precise, single-step guidance requires aligning with the true mean velocity $u(t)$ across the entire frequency spectrum.

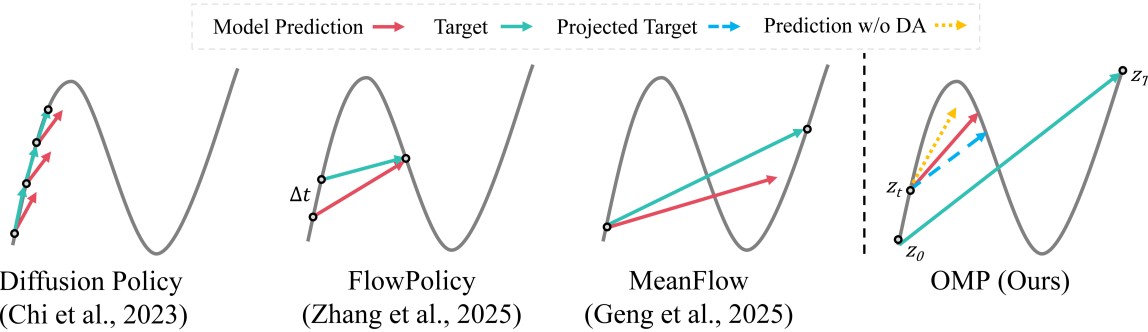

*Figure 3.* **Schematic Comparison of Generative Policy Trajectories.** This diagram contrasts the denoising processes of mainstream paradigms. DP3 requires multi-step denoising (NFE=10). FlowPolicy uses segmented straight-line flows but requires consistency constraints. Mean Policy predicts interval-averaged velocity but may suffer from misalignment between the predicted and target velocities due to training discrepancies. Our proposed OMP introduces Directional Alignment, forcing the predicted velocity vector to align explicitly with the true mean velocity direction, ensuring trajectory accuracy in a single step.

### 4.2.2. GRADIENT STARVATION IN HIGH-DIMENSIONAL GEOMETRY

The second impediment lies in the gradient dynamics of the MSE loss function, specifically regarding directional learning in low-velocity regimes.

Let the predicted velocity be $u = \rho\hat{n}$ and the target be $u^* = \rho^*\hat{n}^*$, where $\rho, \rho^*$ are magnitudes and $\hat{n}, \hat{n}^*$ are unit direction vectors. The MSE loss $\mathcal{L}_{MSE} = \|u - u^*\|_2^2$ can be decomposed using the Law of Cosines:

$$\mathcal{L}_{MSE} = \rho^2 + (\rho^*)^2 - 2\rho\rho^* \cos\alpha, \qquad (8)$$

where $\alpha$ is the angle between $u$ and $u^*$. To understand the learning dynamics, we examine the gradient with respect to the angular error $\alpha$:

$$\frac{\partial \mathcal{L}_{MSE}}{\partial \alpha} = 2\rho\rho^* \sin\alpha. \qquad (9)$$

This equation exposes a fundamental pathology which we term *Gradient Starvation*. The magnitude of the gradient responsible for correcting the direction ($\alpha$) is multiplicatively coupled to the target magnitude $\rho^*$.

In robotic manipulation, the action space dimensionality is substantially lower than in image generation, resulting in the case that $\rho^* \approx 0$. As $\rho^* \to 0$, the angular gradient $\frac{\partial \mathcal{L}}{\partial \alpha} \to 0$. This implies that for those cases, the MSE loss provides effectively zero supervision for directional alignment. The network minimizes loss by simply reducing its output magnitude $\rho$ to zero, rather than aligning $\hat{n}$ with $\hat{n}^*$. This results in a "stationary" policy that fails to execute the necessary directional corrections for successful assembly.

To empirically validate this phenomenon, we conducted a preliminary experiment visualizing the optimization dynamics of a learnable 2D velocity vector, as illustrated in Figure 2(b). We compared the regression performance against two distinct targets: a high-magnitude velocity ($\rho^* = 1.0$) and a low-magnitude velocity ($\rho^* = 0.01$). Consistent with the analytical derivation in Equation 9, the high-magnitude target (blue trajectories) induces a strong directional gradient, enabling the model to simultaneously correct magnitude and direction, resulting in rapid angular convergence. Conversely, the low-magnitude target (orange trajectories) demonstrates the effects of gradient starvation. The optimization path reveals a pathological behavior where the model prioritizes collapsing the magnitude $\rho$ towards zero before attempting to correct the orientation $\alpha$. This is corroborated by the angular error plot, which shows a significant lag in directional learning for the low-velocity target, confirming that standard MSE loss is ill-suited for the precise, low-speed adjustments required in fine manipulation tasks.

### 4.2.3. MEMORY COMPLEXITY OF NESTED DERIVATIVES

The third barrier is the computational burden . The Mean-Flow Identity (Eq. 4) requires computing the total derivative $\frac{d}{dt}u$. Expanding this total derivative via the chain rule yields:

$$\frac{d}{dt}u(z_t, r, t) = \frac{\partial u}{\partial t} + \nabla_z u(z_t, r, t) \cdot \frac{dz}{dt}. \qquad (10)$$

The term $\nabla_z u \cdot \frac{dz}{dt}$ represents a Jacobian-Vector Product (JVP). In a standard training loop, we must compute the gradient of the loss with respect to the parameters $\theta$:

$$\nabla_\theta \mathcal{L} \propto \nabla_\theta \left( \nabla_z u_\theta \cdot v \right). \qquad (11)$$

This effectively requires computing second-order mixed partial derivatives $\frac{\partial^2 u}{\partial\theta\partial z}$. In automatic differentiation (AD) frameworks (e.g., PyTorch, JAX), differentiating a JVP operation requires nesting *Reverse-Mode AD* (for the loss) over *Forward-Mode AD* (for the JVP).

This nesting prevents the standard memory optimization of discarding intermediate activations. The computational

graph must store: primal activations $X$, tangent vectors $\delta X$ (for the inner forward pass), and adjoint gradients of the tangents (for the outer backward pass). This results in a much higher memory complexity than standard backpropagation. For high-dimensional point-cloud encoders (e.g., PointNet++ or Transformers), this memory overhead makes training with sufficient batch sizes or temporal horizons on standard GPUs infeasible.

### 4.3. One-step MeanFlow Policy (OMP)

To address the limitations identified above—specifically the gradient starvation in low-velocity regimes and the memory bottleneck of exact differentiation—we propose the One-step MeanFlow Policy (OMP).

#### 4.3.1. DIRECTIONAL ALIGNMENT LEARNING

To resolve the spectral bias and gradient starvation identified in Section 4.2, we introduce a Directional Alignment mechanism. Unlike MSE, which couples magnitude and direction, we explicitly decouple these components to enforce alignment between the predicted velocity $u(z_t, r, t|c)$ and the true mean velocity $v_0$.

**Directional Alignment.** We formulate the Directional Alignment $\mathcal{L}_{DA}$ to maximize the cosine similarity between the prediction and the ground truth:

$$\cos \alpha = \frac{v_0 \cdot u(z_t, r, t|c)}{||v_0|| \cdot ||u(z_t, r, t|c)||}, \mathcal{L}_{DA} = -\log\left(\frac{\cos \alpha + 1}{2}\right). \quad (12)$$

This objective ensures a non-vanishing directional correction gradient as $||v_0|| \to 0$. In practice, a small constant $\epsilon_{\text{dir}} \approx 10^{-6}$ is added to $||v_0||$ for numerical stability. Crucially, by directly aligning the predicted mean velocity with the true mean velocity $v_0$, we circumvent the spectral bias identified in Section 4.2.1.

**Composite Loss Function.** The final OMP training objective combines the reconstruction, dispersive, and alignment terms:

$$\mathcal{L} = \mathcal{L}_{mse} + \lambda_{Disp} \cdot \mathcal{L}_{Disp} + \lambda_{DA} \cdot \mathcal{L}_{DA}. \quad (13)$$

This formulation ensures robust learning across both ballistic transit motions (dominated by $\mathcal{L}_{mse}$) and high-precision contact phases (dominated by $\mathcal{L}_{DA}$).

#### 4.3.2. EFFICIENT OPTIMIZATION VIA DIFFERENTIAL DERIVATION

To overcome the memory prohibitive costs of the exact JVP operator detailed in Section 4.2.3, we implement a Differential Derivation Equation (DDE). Motivated by Wang et al. (2025c), we approximate the time derivative of the network output using a finite difference method rather than symbolic differentiation.

**The DDE Approximation.** We replace the analytical derivative in the MeanFlow Identity with:

$$\frac{du_\theta(z_t, t, r|c)}{dt} \approx \frac{u_\theta(z_{t+\epsilon}, t+\epsilon, r|c) - u_\theta(z_{t-\epsilon}, t-\epsilon, r|c)}{2\epsilon}, \quad (14)$$

where $\epsilon$ is a perturbation constant. This approximation decouples the forward and backward passes, obviating the need to store the intermediate Jacobian computational graph. This reduction in memory overhead allows for the training of deeper networks and longer context windows without sacrificing prediction performance.

## 5. Experimental Results

### 5.1. Experimental Setup

**Dataset and Tasks.** As shown in Figure 4, we evaluate our proposed method on multiple tasks, including 3 from the Adroit benchmark, 34 tasks from the Meta-World benchmark and 4 tasks in real-world setting. The tasks from the Meta-World Benchmark can be classified into twenty-one Easy tasks, four Medium tasks, four Hard tasks and five Very Hard tasks. For the real-word experiments, we design 3 robot manipulation tasks, including Place Bottle, Clean Table and Slip Ring.

**Baselines.** We compare our proposed OMP with multiple SOTA Baselines, including DP, AdaFlow, and CP with 2D inputs, as well as DP3, Simple DP3, FlowPolicy, and MP1 with 3D inputs. DP, DP3, and Simple DP3 all employ multistep inference (NFE=10), whereas CP and FlowPolicy use single-step inference (NFE=1), and AdaFlow operates with a variable NFE.

**Training Details.** To train on Adroit and Meta-World tasks, we generate 10 expert demonstrations for each simulation task. When dealing with point-cloud data, we use the Farthest Point Sampling (FPS) strategy to reduce the number of points to either 512 or 1024, while for image data, we downsample the resolution to $84 \times 84$ pixels. All evaluated baselines and OMP are assessed through three experiments, each corresponding to a different random seed (0, 10, and 20), and in each experiment, the evaluated method is required to train for 3000 epochs per task on Adroit and 1000 epochs per task on Meta-World. Performance is evaluated every 200 epochs, with final performance calculated as the average of the five highest success rates, and the overall success rate and standard deviation for each task are computed across all three experiments. All training and testing procedures are performed on an NVIDIA RTX 4090 GPU with a batch size of 128, and for optimization, we use the AdamW optimizer with a learning rate of 0.0001 (consistently applied to both Adroit and Meta-World), along with an observation history of 2 steps, a prediction horizon of 4

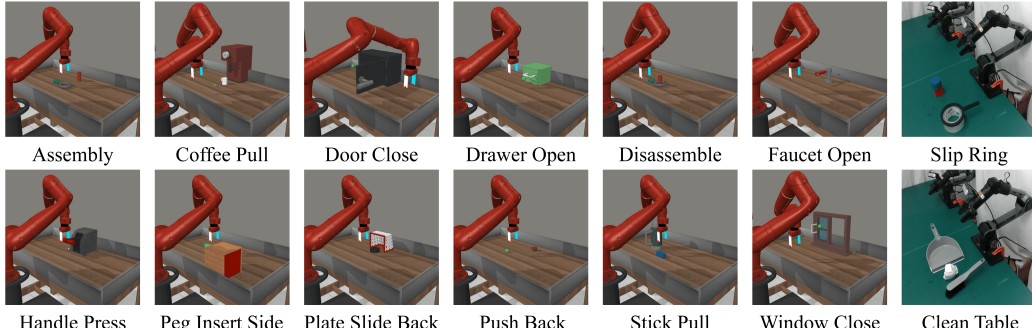

Figure 4. **Overview of Experimental Environments.** Visualizations of the diverse manipulation tasks used for evaluation. The first six columns illustrate the Simulation Benchmark Environments, while the rightmost column depicts the Real-World Robot Experiments.

| Methods | NFE | Adroit | | | Meta-World | | | | Average |
| --- | --- | --- | --- | --- | --- | --- | --- | --- | --- |
| | | Hammer | Door | Pen | Easy (21) | Medium (4) | Hard (4) | Very Hard (5) | |
| DP(RSS'23) | 10 | 16±10 | 34±11 | 13±2 | 50.7±6.1 | 11.0±2.5 | 5.25±2.5 | 22.0±5.0 | 35.2±5.3 |
| Adaflow(NeuRIPS'24) | - | 45±11 | 27±6 | 18±6 | 49.4±6.8 | 12.0±5.0 | 5.75±4.0 | 24.0±4.8 | 35.6±6.1 |
| CP(arxiv'24) | 1 | 45±4 | 31±10 | 13±6 | 69.3±4.2 | 21.2±6.0 | 17.5±3.9 | 30.0±4.9 | 50.1±4.7 |
| DP3(RSS'24) | 10 | **100**±0 | 56±5 | 46±10 | 87.3±2.2 | 44.5±8.7 | 32.7±7.7 | 39.4±9.0 | 68.7±4.7 |
| Simple DP3(RSS'24) | 10 | 98±2 | 40±17 | 36±4 | 86.8±2.3 | 42.0±6.5 | 38.7±7.5 | 35.0±11.6 | 67.4±5.0 |
| FlowPolicy(AAAI'25) | 1 | 98±1 | 61±2 | 54±4 | 84.8±2.2 | 58.2±7.9 | 40.2±4.5 | 52.2±5.0 | 71.6±3.5 |
| MP1(AAAI'26) | 1 | **100**±0 | **69**±2 | 58±5 | 88.2±1.1 | 68.0±3.1 | 58.1±5.0 | 67.2±2.7 | 78.9±2.1 |
| OMP-JVP (Proposed) | 1 | **100**±0 | 68±3 | 60±4 | **89.7**±0.7 | **77.4**±2.2 | **62.5**±3.1 | **77.8**±3.0 | **82.3**±1.6 |
| $-\mathcal{L}_{DA}$ | 1 | **100**±0 | **69**±2 | 58±5 | 88.2±1.1 | 68.0±3.1 | 58.1±5.0 | 67.2±2.7 | 78.9±2.1 |
| $-\mathcal{L}_{dis}$ | 1 | **100**±0 | **69**±2 | 58±4 | 89.1±1.0 | 76.8±3.1 | 57.8±4.2 | 75.0±3.7 | 81.2±2.2 |
| $-\mathcal{L}_{dis} - \mathcal{L}_{DA}$ | 1 | 95±5 | 66±3 | 48±4 | 87.5±1.8 | 73.0±2.7 | 57.5±4.5 | 68.0±3.5 | 78.3±2.6 |
| OMP-DDE (Proposed) | 1 | **100**±0 | 68±2 | **64**±3 | 89.0±1.3 | 76.4±2.7 | 61.0±3.0 | 70.6±4.9 | 80.8±2.2 |
| $-\mathcal{L}_{DA}$ | 1 | 96±4 | 61±2 | 59±4 | 86.1±1.5 | 68.1±3.0 | 56.9±4.3 | 66.7±3.7 | 77.2±2.4 |
| $-\mathcal{L}_{dis}$ | 1 | **100**±0 | 67±2 | 57±4 | 88.4±1.2 | 75.6±2.6 | 57.6±3.8 | 70.4±3.9 | 80.1±2.1 |
| $-\mathcal{L}_{dis} - \mathcal{L}_{DA}$ | 1 | 95±5 | 60±3 | 55±5 | 85.9±1.6 | 66.2±2.9 | 55.7±4.0 | 65.3±4.1 | 76.4±2.6 |

Table 1. **Main Results and Ablation Study on Adroit and Meta-World Benchmarks.** A comprehensive performance comparison across 37 tasks using 3 random seeds. **NFE** denotes the Number of Function Evaluations required for inference (lower is faster).

steps, and an execution horizon of 3 steps. See more details in Section D.

### 5.2. Performance Analysis

**Quantitative Evaluations on Simulation Environments.** Table 1 demonstrates that our proposed OMP achieves superior performance compared to SOTA baselines. OMP achieves an average performance of $82.3\% \pm 1.7\%$, outperforms the latest SOTA method MP1, which achieves an average performance of $78.9\% \pm 2.1\%$. OMP yields a $3.4\%$ improvement over MP1 and a $10.7\%$ improvement over Flowpolicy. The improvement of the average performance over MP1 is not quite significant since MP1 achieves a near-optimal performance, i.e., $88.2\% \pm 1.1\%$ success rate, on Meta-World easy tasks, which contains the majority of simulation tasks, i.e., 21 tasks over totally 37 tasks. While we focus our attention on some poorly-performed tasks, like Meta-World Medium, Hard, and Very Hard tasks, our improvement over MP1 is significant. OMP yields a $9.4\%$ improvement over Meta-World Medium tasks, a $4.4\%$ improvement over Meta-World Hard tasks and a $10.6\%$ improvement over Meta-World Very Hard tasks. Meanwhile,

the DDE version, OMP-DDE, also yields a $8.4\%$ improvement over Meta-World Medium tasks, a $2.9\%$ improvement over Meta-World Hard tasks and a $3.4\%$ improvement over Meta-World Very Hard tasks.

| Methods | Place | Clean | Slip |
| --- | --- | --- | --- |
| DP3(RSS'24) | 65.0% | 60.0% | 50.0% |
| FlowPolicy(AAAI'25) | 60.0% | 50.0% | 40.0% |
| MP1(AAAI'26) | 70.0% | 65.0% | 55.0% |
| OMP(Proposed) | **80.0%** | **75.0%** | **70.0%** |

Table 2. Quantitative comparison of success rates across three real-world manipulation tasks (Place Bottle, Clean Table, Slip Ring).

**Quantitative Evaluations on Real-world Tasks.** Table 2 presents the quantitative comparison of success rates across three real-world manipulation tasks: Place Bottle, Clean Table, and Slip Ring. The proposed method consistently outperforms all state-of-the-art baselines (DP3, FlowPolicy, and MP1) across every evaluated scenario. Specifically, our approach achieves the highest success rates of $80.0\%$, $75.0\%$, and $70.0\%$ for the Place, Clean, and Slip tasks, respectively. This corresponds to a significant performance gain over the strongest baseline, MP1, with improvements of $10\%$ in the Place and Clean tasks and a substantial $15\%$

increase in the more challenging Slip Ring task. While baseline methods struggle significantly with the Slip task, achieving success rates between $40.0\%$ and $55.0\%$, OMP demonstrates superior robustness and capability in handling complex real-world dynamic manipulation challenges.

**JVP vs. DDE.** The comparison between OMP-JVP and OMP-DDE reveals a clear trade-off between calculation precision and memory efficiency. As detailed in Table 1, replacing the exact JVP operator with the DDE approximation introduces some calculation errors, resulting in a moderate performance drop. Specifically, the average success rate decreases from $82.3 \pm 1.7\%$ with OMP-JVP to $80.8 \pm 2.2\%$ with OMP-DDE, with more pronounced deficits in complex scenarios like the Meta-World "Very Hard" tasks ($77.8\%$ vs. $70.6\%$).

| Tasks | OMP-JVP | OMP-DDE |
|---|---|---|
| Adroit Hammer(Horizon=4) | 6.60GB | 5.35GB |
| Place Bottle(Horizon=4) | 23.49GB | 18.33GB |
| Adroit Hammer(Horizon=16) | 7.69GB | 6.12GB |
| Place Bottle(Horizon=16) | 26.71GB | 19.19GB |

*Table 3.* GPU memory usage (GB) comparison on different tasks with different lengths of action chunk during training.

However, this slight reduction in accuracy is justified by the substantial reduction in computational cost shown in Table 3. The DDE method consistently lowers GPU memory usage; for example, in the "Place Bottle" task (Horizon= 16), memory consumption drops significantly from 26.71GB to 19.19GB. This efficiency gain is highly sensitive to input size: the experiments used relatively small point clouds, $(512, 3)$ for Adroit Hammer and $(1024, 3)$ for Place Bottle, yet the larger "Place Bottle" input already demonstrates a much wider memory gap (saving $\approx 7.5$GB) compared to the Adroit task. In real-world applications requiring high-fidelity point clouds or high-resolution images, the memory overhead of the standard JVP would likely become prohibitive, making the DDE approximation a crucial implementation strategy despite the marginal performance cost.

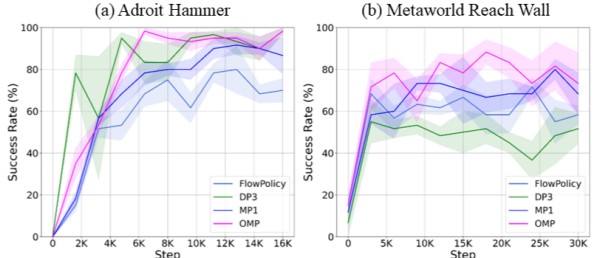

*Figure 5.* **Training Stability and Convergence Analysis.** Success rate curves during training for (a) Adroit Hammer and (b) Meta-World Reach Wall.

**Training Efficiency.** Figure 5 illustrates the training-phase success rate curves of baseline methods (FlowPolicy, DP3, MP1) and our proposed OMP. While baseline methods exhibit significant volatility—most notably DP3, which performs competitively on Adroit Hammer but degrades largely on Metaworld Reach Wall—OMP maintains a consistently high and stable training trajectory across both tasks. By contrast, OMP achieves rapid convergence and superior final success rates compared to all baselines, a benefit attributed to its additional guidance mechanism. We have also overlaid a shadow area on each curve, representing the standard deviation across different random seeds. This visualization further underscores the superior stability of OMP, which displays markedly tighter variance compared to the wide fluctuations observed in baselines such as FlowPolicy.

**Ablation Study.** We ablate the Dispersive Loss ($\mathcal{L}_{dis}$) and Directional Alignment ($\mathcal{L}_{DA}$) to isolate their contributions to both the JVP and DDE variants of OMP (Table 1). Removing the Dispersive Loss ($-\mathcal{L}_{dis}$) yields a minor decline in overall average success ($82.3\%$ to $81.2\%$ for JVP; $80.8\%$ to $80.1\%$ for DDE). While this decline is relatively minor on easier tasks, it becomes more noticeable on complex scenarios, such as the Meta-World Hard tasks. Conversely, Directional Alignment is critical to baseline performance. Removing it entirely ($-\mathcal{L}_{DA}$) causes a substantial drop in average success across both variants (falling to $78.9\%$ for JVP and $77.2\%$ for DDE). Eliminating both components ($-\mathcal{L}_{dis} - \mathcal{L}_{DA}$) degrades performance further to $78.3\%$ and $76.4\%$, respectively. This degradation is exceptionally severe in dexterity-heavy environments; for instance, the success rate on Adroit Pen falls from $60\%$ to $48\%$ (JVP) and $64\%$ to $55\%$ (DDE) when both losses are removed. These results confirm that $\mathcal{L}_{DA}$ provides the fundamental guidance necessary for stable, high-fidelity single-step generation. Further sensitivity analyses on horizon length and DDE parameter $\epsilon$ are provided in Section E.1 and Section E.2.

# 6. Conclusion

In this paper, we introduce OMP, a novel framework designed to bridge the gap between high-fidelity generation and real-time inference in robotic manipulation. We identify and analyze critical theoretical pathologies inherent in applying the MeanFlow paradigm to robotics, specifically Numerical Instability in Short-Horizon Denoising and Gradient Starvation in low-velocity regimes. To resolve these, we propose a lightweight Directional Alignment mechanism that explicitly synchronizes predicted velocities with true mean velocities, ensuring robust trajectory learning. Furthermore, to mitigate the high memory complexity of exact Jacobian-Vector Product computations, we implement a Differential Derivation Equation (DDE). Extensive experiments on Adroit, Meta-World, and real-world setups demonstrate

that OMP outperforms state-of-the-art baselines like MP1 and FlowPolicy in both success rate and training stability, establishing a new standard for efficient, single-step generative policy learning.

## Acknowledgments

This work has been supported in part by the program of National Natural Science Foundation of China (No. 62176154), the program of National Natural Science Foundation of China (No. 62503322), the AI for Science Seed Program of Shanghai Jiao Tong University (project number 2025AI4SQY06), the Shanghai Jiao Tong University Xiaomi Scholar Fund, and the Shanghai Municipal Special Program for Basic Research on General AI Foundation Models.

## Impact Statement

This paper introduces a novel framework for efficient, high-fidelity robotic manipulation, specifically targeting the latency and computational bottlenecks of generative policy learning. While the primary goal is to advance the technical capabilities of embodied AI, enabling real-time control for applications ranging from industrial assembly to household assistance, we acknowledge the broader societal implications inherent to automation. The advancement of robust, high-frequency robot control contributes to the ongoing dialogue regarding labor displacement and the dual-use nature of autonomous systems; however, our work focuses on foundational algorithmic improvements and does not introduce specific ethical concerns regarding data privacy, bias, or surveillance beyond those established in the general field of machine learning.

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

# A. Additional Related Works

In this section, we contextualize our approach within the broader landscape of robotic learning. The following discussion covers three distinct streams of research: reinforcement learning, foundation models, and world models.

## A.1. Reinforcement Learning for Robotic Manipulation

Recent advancements in reinforcement learning for robotic manipulation have increasingly focused on integrating semantic understanding, improving algorithmic efficiency, and refining online adaptation mechanisms. Leveraging natural language to guide policy learning, Zhang et al. (2025a) utilize language-guided rewards to train policies without requiring new demonstrations, while Chowdhury et al. (2025) introduce LAGEA to direct embodied agents in complex manipulation tasks. To address sample efficiency and stability in high-dimensional control, Jiang et al. (2025c) incorporate time reversal symmetry into the learning process, and Seo et al. (2025) propose FastTD3, a streamlined algorithm capable of robust humanoid control.

Furthermore, novel strategies have been developed to enhance policy adaptability and safety. Zhang et al. (2025b) fine-tune flow matching policies using online reinforcement learning, while Murillo-Gonzalez & Liu (2025) tackle the related challenge of continual robot learning through Action Flow Matching. Emphasizing reliability, Dai et al. (2025) integrate safety constraints directly into motion planning with SafeFlow, complementing works like Lu et al. (2025c) that employ human-in-the-loop online rejection sampling.

## A.2. Foundation Models and Generative Architectures

Recent research has focused on scaling robot learning through generalist foundation models and advanced generative architectures. Kim et al. (2024) and Team et al. (2024) establish open-source benchmarks with OpenVLA and Octo, respectively, providing widely adaptable policies trained on massive cross-embodiment datasets.

Generative approaches, particularly Flow Matching and Diffusion, have become central to this domain. In diffusion, Liu et al. (2024b) propose RDT-1B for bimanual manipulation, while Vosylius & Johns (2025) utilize graph diffusion for in-context refinement. Flow matching has seen rapid expansion following $\pi_0$ (Black et al., 2026). Recent works like VITA (Gao et al., 2025) and AsyncVLA (Jiang et al., 2025b) adapt this paradigm specifically for vision-language-action tasks. Others focus on structural simplification and modality: Jiang et al. (2025a) treat action trajectories as flow streams, Noh et al. (2025) extend the generation to 3D space, and methods like InstaFlow (Liu et al., 2024c) and AdaFlow (Hu et al., 2024) optimize for inference efficiency.

Finally, to address the computational constraints of Transformers, Mamba-based architectures have gained traction. Liu et al. (2024a) present RoboMamba, an efficient Vision-Language-Action model, while Wang et al. (2025b) introduce FlowRAM, which grounds flow matching policies within a Region-Aware Mamba framework to enhance spatial reasoning.

## A.3. World Models for Robotics

Generative world models are increasingly used to simulate physics and predict future states for robotic planning. In the domain of manipulation, Lu et al. (2025b) leverage Gaussian representations to construct scalable world models, while Jiang et al. (2025d) utilize diffusion world models to explicitly refine policies for reinforcement learning. Guo et al. (2025) introduce Ctrl-World to enable controllable video generation for downstream decision-making tasks. Expanding into dynamic 4D environments, Mao et al. (2025) propose DreamDrive, which generates consistent 4D scene models from street-view imagery to facilitate realistic simulation.

# B. Detailed Analysis on Meanflow Framework

In this section, we provide a detailed mathematical analysis to the three barriers we proposed in Section 4.2.

## B.1. The Averaging Operator as a Low-Pass Filter

To analyze the spectral properties of MeanFlow, we must consider the relationship between the instantaneous velocity $v(t)$ and the average velocity $u(t)$ in the frequency domain. Let the ground truth trajectory be a continuous function $x(t)$ for $t \in [0, 1]$. The instantaneous velocity is the time derivative $v(t) = \dot{x}(t)$.

The MeanFlow model learns the average velocity $u(t)$, which is defined as the integral of the instantaneous velocity normalized by the time interval:

$$u(t) = \frac{1}{t} \int_0^t v(\tau) d\tau \tag{15}$$

The error in the generated sample $\hat{x}_0$ relative to the true $x_0$ is given by $\epsilon_{gen} = t\|\hat{u}(t) - u(t)\|$. This suggests that minimizing the error in $u$ directly minimizes the generation error. However, we assert that $u(t)$ is inherently a low-frequency signal compared to $v(t)$.

**Theorem B.1** (Spectral Decay of the Average Velocity Field). *Let $v(t)$ be a wide-sense stationary stochastic process with power spectral density $S_v(\omega)$. Let $u(t)$ be the average velocity process defined above. Then, for large frequencies $\omega$, the power spectral density of the average velocity, $S_u(\omega)$, decays at a rate of $O(1/\omega^2)$ relative to $S_v(\omega)$.*

*Proof.* We analyze the operation in the Fourier domain. Let $\mathcal{F}[f](\omega) = \hat{F}(\omega)$ denote the Fourier transform of a function $f(t)$. Since $v(t) = \dot{x}(t)$, we have the relationship:

$$\hat{V}(\omega) = i\omega \hat{X}(\omega) \implies \hat{X}(\omega) = \frac{\hat{V}(\omega)}{i\omega} \tag{16}$$

The average velocity $u(t)$ can be expressed in terms of the position as $u(t) = \frac{x(t)-x(0)}{t}$. For $\omega > 0$, the spectral content of the numerator $\Delta x(t)$ is identical to that of $x(t)$. The dominant effect of the operation is the integration. Thus, the magnitude of the spectral components relates as:

$$|\hat{U}(\omega)| \propto |\hat{X}(\omega)| = \left| \frac{\hat{V}(\omega)}{i\omega} \right| = \frac{|\hat{V}(\omega)|}{|\omega|} \tag{17}$$

The Power Spectral Density (PSD) is proportional to the square of the magnitude of the Fourier transform. Therefore:

$$S_u(\omega) \propto |\hat{U}(\omega)|^2 \propto \frac{|\hat{V}(\omega)|^2}{\omega^2} = \frac{S_v(\omega)}{\omega^2} \tag{18}$$

$\square$

This result proves that the averaging operator acts as a first-order low-pass filter (integrator). Every doubling of frequency in the input signal $v(t)$ results in a 6 dB reduction in the power of that frequency in the target signal $u(t)$, in addition to whatever spectral decay $S_v(\omega)$ already possesses.

## B.2. Gradient Starvation in High-Dimensional Geometry

We provide a rigorous basis for the gradient starvation by decomposing the gradient vector field into radial and tangential components, proving that the optimization dynamics inherently favor magnitude collapse over directional alignment.

To analyze the trajectory of learning, we project the gradient $\nabla_u \mathcal{L}$ onto the polar basis $\{\hat{e}_\rho, \hat{e}_\tau\}$. The gradient vector is given by:

$$\nabla_u \mathcal{L} = 2(\rho - \rho^* \cos\alpha)\hat{e}_\rho + 2\rho^* \sin\alpha \hat{e}_\tau. \tag{19}$$

We define the *Directional Stiffness Ratio* $\gamma$ as the ratio of the tangential (corrective) force to the radial (magnitude) force. In the fine manipulation regime where $\rho^* \ll \rho$:

$$\gamma = \frac{\|\text{Tangential Component}\|}{\|\text{Radial Component}\|} \approx \frac{\rho^* |\sin\alpha|}{\rho}. \tag{20}$$

**Angular Freezing.** Considering continuous-time gradient descent dynamics, the angular velocity of the parameters $\dot{\alpha}$ relates to the target magnitude $\rho^*$ as:

$$\dot{\alpha} \propto \rho^* \sin\alpha. \tag{21}$$

As $\rho^* \to 0$, $\dot{\alpha} \to 0$, while the magnitude dynamics $\dot{\rho} \approx -2\rho$ remain active. Consequently, the optimization path moves strictly along $-\hat{e}_\rho$, collapsing $\rho \to 0$ before the angle $\alpha$ can rotate, resulting in the observed stationary policy.

## B.3. Memory Complexity of Nested Derivatives

We quantify the memory bottleneck by analyzing the space complexity of the augmented computational graph required for second-order optimization, demonstrating that the memory footprint scales linearly with the tangent dimensionality.

**Memory Doubling.**    Let the neural network be represented as a composition of $L$ layers, where the activation at layer $l$ is $h_l = f_l(h_{l-1}; \theta)$. In standard training (first-order optimization), the backpropagation algorithm requires storing the primal activations $\{h_l\}_{l=1}^{L}$ to compute gradients. The memory complexity $\mathcal{M}_{std}$ is proportional to the sum of feature volumes:

$$\mathcal{M}_{std} \propto \sum_{l=1}^{L} \text{size}(h_l). \tag{22}$$

When computing the total derivative via the JVP, we effectively forward-propagate a tangent vector $\delta h$ alongside the primal state. The operation at layer $l$ becomes an augmented transformation $F_l$:

$$(h_l, \delta h_l) = F_l(h_{l-1}, \delta h_{l-1}) = \left( f_l(h_{l-1}), \frac{\partial f_l}{\partial h_{l-1}} \delta h_{l-1} \right). \tag{23}$$

To train the parameters $\theta$ based on JVP, we must perform reverse-mode differentiation on this augmented graph. This requires storing the inputs to $F_l$ for the backward pass. The storage requirement becomes:

$$\mathcal{M}_{JVP} \propto \sum_{l=1}^{L} \left( \text{size}(h_l) + \text{size}(\delta h_l) \right). \tag{24}$$

Since the tangent vector $\delta h_l$ shares the same dimensionality as the primal state $h_l$ (e.g., $B \times N \times D$ for point clouds), we obtain:

$$\mathcal{M}_{JVP} \approx 2 \cdot \mathcal{M}_{std}. \tag{25}$$

This derivation proves that training on the total derivative doubles the activation memory requirement per layer. For memory-bound architectures like PointNet++ (where $N$ is large), this $2\times$ factor forces a reduction in batch size $B$ or temporal horizon $T$ by half, often rendering the training of long-horizon physics models computationally intractable on standard hardware.

# C. Preliminary Experiments

## C.1. Preliminary Experiments Setup

### C.1.1. FREQUENCY-SPECIFIC CONVERGENCE RATE

**"Sum of Sines" Signal Generator.**    Instead of using complex real-world data (like images or robot trajectories), we defined a synthetic ground-truth velocity field $v(t)$ composed of pure sine waves at specific frequencies (e.g., 1 Hz, 8 Hz). This allowed us to precisely control the "stiffness" of the dynamics. For every training sample, we randomized the phase of these waves. This forced the neural network to actually learn the function mapping time and phase to velocity, rather than simply memorizing a fixed trajectory.

**Dual-Target Training Regime.**    We trained two identical Multi-Layer Perceptrons (MLPs) on this data to act as a side-by-side comparison. The Baseline (Flow Matching): This network attempted to predict the instantaneous velocity $v(t)$ directly. The Experiment (MeanFlow): This network attempted to predict the time-averaged velocity $u(t) = \frac{1}{t-r} \int_r^t v(\tau) d\tau$. Crucially, we computed this target integral analytically in the data loader. By feeding the network the exact mathematical integral of the sines, we ensured that any failure to learn was due to the network's optimization dynamics (the spectral bias), not numerical errors in creating the target.

**"Inverse Transform" Evaluation.**    To compare the two models fairly, we could not simply compare their loss functions, as they were minimizing different quantities ($v$ vs $u$). We implemented a validation step using Automatic Differentiation. For the MeanFlow model, we took its output $\hat{u}(t)$ and explicitly calculated its derivative with respect to time $\partial \hat{u}/\partial t$ to recover the implied instantaneous velocity using the MeanFlow Identity: $\hat{v} = \hat{u} + (t - r)\dot{\hat{u}}$. This allowed us to plot the error of both models in the same "velocity space" (as seen in the figures), revealing exactly how the averaging operator filtered out the high-frequency 8 Hz and 64 Hz signals.

### C.1.2. ANGULAR ERROR COMPARISON BETWEEN TARGETS WITH DIFFERENT MAGNITUDE

To rigorously investigate the hypothesis of gradient starvation, we designed a controlled optimization experiment that decouples directional alignment from magnitude regression. This setup isolates the geometric properties of the loss function by removing the confounding variables of network capacity and deep layer propagation. The experimental domain consists of a 2D point navigation task where the agent is positioned at the origin. We defined a deterministic mapping from random input state vectors to a specific target direction angle, effectively creating a "needle in a haystack" search problem where the correct direction must be located without magnitude cues.

Crucially, to analyze the optimization path, we initialized the output vector with a substantial magnitude along a fixed axis, intentionally misaligned with the target direction. Rather than updating the network parameters, we froze the weights and performed gradient descent directly on the output vector with respect to the standard Mean Squared Error (MSE) objective. This procedure allows for the direct observation of the optimization trajectory in the magnitude-phase space over 1,000 steps. We quantify the optimization dynamics using a "Starvation Ratio," which compares the relative rate of magnitude reduction to the rate of angular correction. A ratio significantly greater than one serves as empirical evidence that the optimizer prioritizes collapsing the vector magnitude towards stationarity—the path of least resistance—rather than correcting the directional alignment.

## C.2. Additional Preliminary Experimental Results

### C.2.1. ADDITIONAL PRELIMINARY EXPERIMENTS ON SPECTRAL BIAS

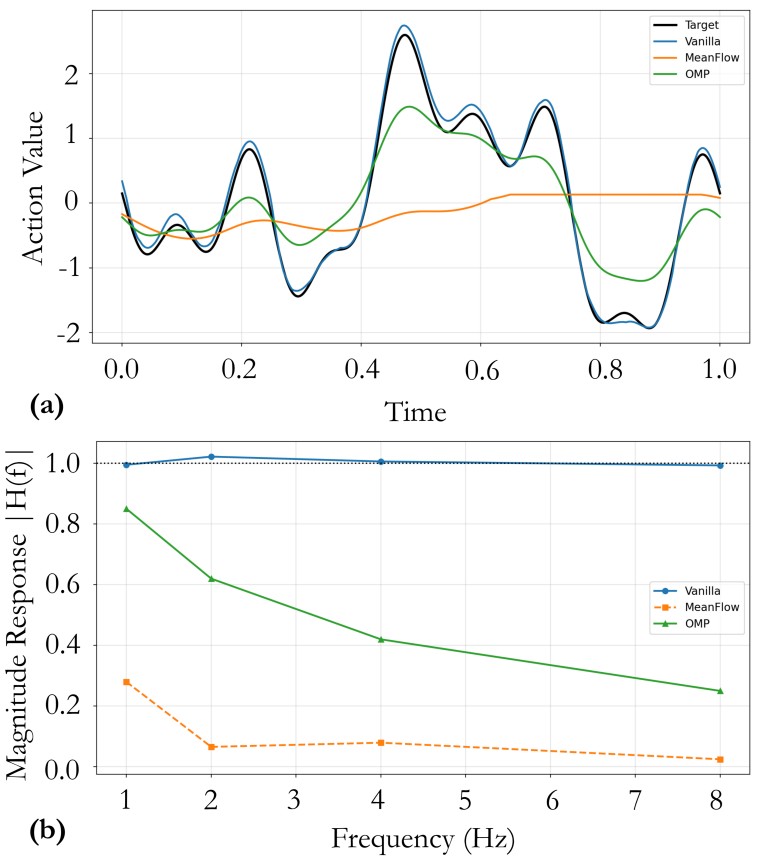

*Figure 6.* Multi-scale signal reconstruction comparison. (a) Target and reconstructed signals (Vanilla, MeanFlow, OMP) in the normalized time domain. (b) Frequency-dependent reconstruction fidelity metric (0 to 1 scale, where 1.0 represents perfect reconstruction).

**Experimental Setup.** Based on the preliminary experiments in Section 4.2.1, we extend our preliminary experiments in a frequency-controlled setting to further exhibit the reaction of different methods to cross-frequency signals. We trained a

Vanilla Multi-step policy, a standard MeanFlow policy, and our proposed OMP using identical network backbones on a composite multi-frequency trajectory:

$$a_{\text{trajectory}}(t) = \sum_{i=1}^{N} A_i \sin(2\pi f_i t + \phi_i), \tag{26}$$

where $N = 4$ and the target frequencies are $f_i \in \{1, 2, 4, 8\}$ Hz. The models are evaluated in both the trajectory domain (Figure 6A) and the frequency domain (Figure 6B). To evaluate the frequency domain, we compute the magnitude response $|H(f)|$ of each policy by taking the ratio of the predicted amplitude to the target amplitude at each frequency bin:

$$|H(f)| = \frac{|\text{FFT}(a_{\text{pred}})|}{|\text{FFT}(a_{\text{trajectory}})|}, \tag{27}$$

where FFT denotes the Fast Fourier Transform and $a_{\text{pred}}$ is the predicted trajectory.

**Results and Analysis.**

- **Vanilla Multi-step:** This baseline perfectly tracks the trajectory and maintains a flat magnitude response ($|H(f)| \approx 1.0$) across all frequencies. This confirms that the network backbone is fully capable of learning high-frequency signals when unconstrained by single-step integration.

- **MeanFlow:** Consistent with the reviewer's hypothesis, standard MeanFlow suffers from training collapse. It exhibits severe spectral attenuation, struggling to capture even the 1 Hz base frequency ($|H(f)| \approx 0.28$). While this general instability heavily obscures the method's underlying structural low-pass properties, the magnitude response (Figure 6B) still demonstrates a greater capacity for low-frequency refinement than high-frequency refinement, aligning with our low-pass filter analysis.

- **OMP (Ours):** By aligning with the true mean velocity, our proposed OMP stabilizes training and captures the 1 Hz base frequency with high fidelity ($|H(f)| \approx 0.85$). Although the magnitude response decays smoothly at 2, 4, and 8 Hz, OMP successfully increases the magnitude response at these higher frequencies compared to standard MeanFlow.

**Conclusion.** Because our stabilized OMP effectively learns the 1 Hz component while predictably attenuating at higher frequencies, this experiment successfully isolates the structural limitation: single-step integration inherently imposes a low-pass filter, independent of training collapse. Furthermore, Figure 6B demonstrates that OMP significantly improves the magnitude response across the entire spectrum relative to standard MeanFlow.

C.2.2. ADDITIONAL PRELIMINARY EXPERIMENTS ON GRADIENT STARVATION

To validate the practical relevance of the gradient starvation analysis, we plot the distribution of normalized velocity magnitudes for 4 representative tasks. The empirical distributions (Figure 7) confirm that small velocities are common in different tasks. While peak velocities vary by task, all exhibit long-tail distributions approaching zero. Notably, tasks like Plate Slide display substantial density at low magnitudes (e.g., below 0.4). These statistics demonstrate that the gradient starvation problem analyzed in Section 4.2.2 is a frequent, practical issue in real-world settings rather than a theoretical edge case.

## D. Detailed Experimental Setup

Table 4 provides a comprehensive overview of the hyperparameters used across all experiments. While the configuration for the Adroit and Meta-World benchmarks was detailed in Section 5.1, we now elaborate on the specific protocols employed for our real-world experiments. Codes are included in our supplementary material.

**Experimental Setup for Real-world Experiments.** Figure 8 provides a visualization of the three real-world tasks used to evaluate our proposed OMP and other baselines. For each task, we collected 50 expert demonstrations. To process the 3D data, we employed Farthest Point Sampling (FPS) to downsample point clouds to 1,024 points. All models, including baselines and OMP, were trained for 3,000 epochs on an NVIDIA RTX 4090 GPU. We utilized a batch size of 128 and the

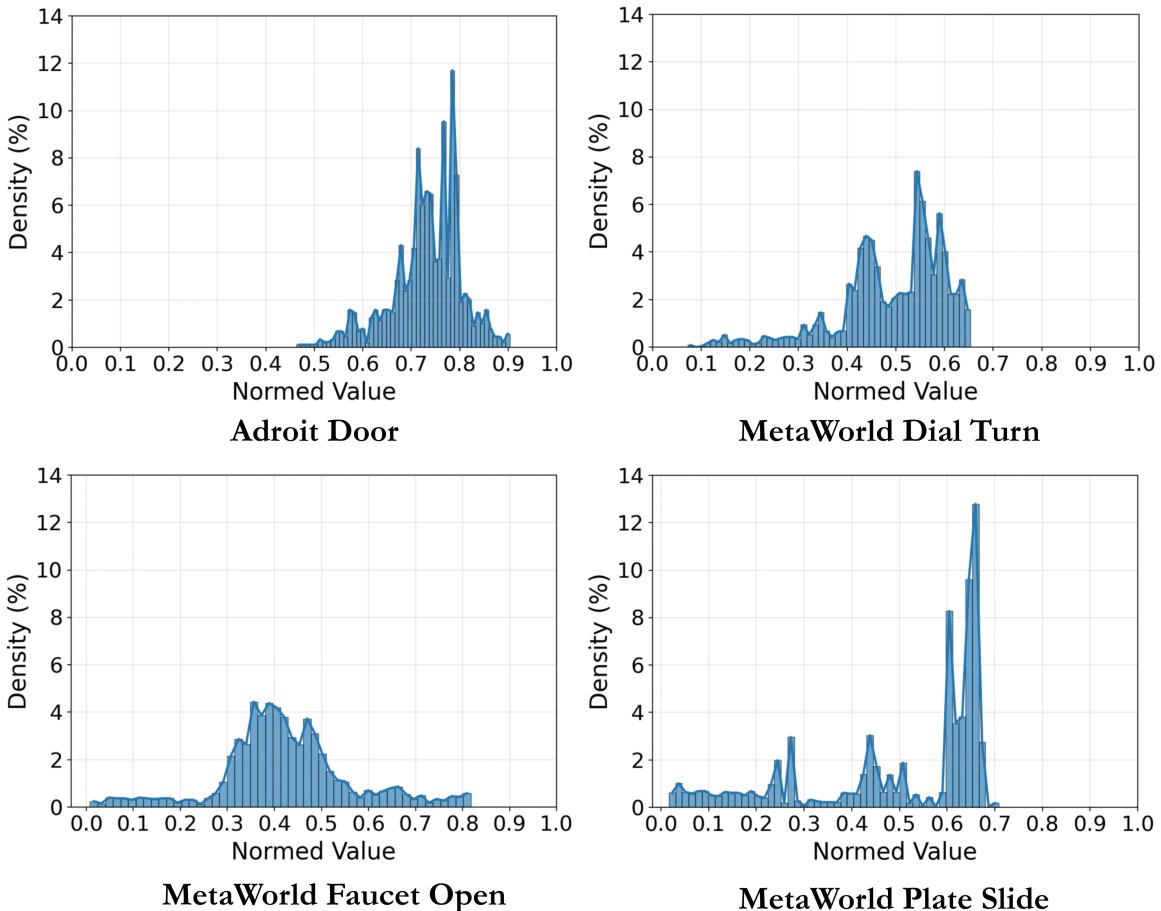

*Figure 7.* Empirical distributions of normed velocities across four representative tasks: Adroit Door, MetaWorld Dial Turn, MetaWorld Faucet Open, and MetaWorld Plate Slide.

AdamW optimizer with a learning rate of $1 \times 10^{-4}$. The policy was configured with an observation window of 2 steps, a prediction horizon of 16 steps, and an execution horizon of 8 steps. Evaluation was conducted over 20 trials with randomized object placement for each method. **Real-world experiment demo can be found in our supplementary materials.**

## E. Additional Experimental Results

### E.1. Sensitivity Analysis on Horizon Length

**Experimental Setup.** We performed a sensitivity analysis to assess how varying action chunk lengths affects performance. Our evaluation compares three configurations—Short, Medium, and Long—defined by prediction horizons of 4, 8, and 16 steps, and execution horizons of 3, 4, and 8 steps, respectively, all maintained with a constant 2-step observation window.

**JVP vs DDE.** The results in Table 5 illustrate the performance trade-offs between the JVP and DDE variants across varying action chunk scales. In the Short and Medium settings, the JVP variant generally outperforms or matches DDE, achieving the highest average success rates of 82.3% in Short and 80.7% in Medium. JVP demonstrates particular strength in the Meta-World benchmarks at these scales, suggesting that its Jacobian-vector product formulation provides superior precision for shorter-horizon control.

However, as the action chunk length increases to the Long setting ($H_p = 16, H_e = 8$), we observe a shift in the relative performance. While JVP maintains a slight edge in the Meta-World Medium tasks, the DDE variant exhibits better robustness in the Pen task (70% vs 64%) and the Meta-World Very Hard tasks (75.2% vs 71.3%), leading to a higher overall average in the Long category (78.2% vs 76.6%). This suggests that while JVP is highly effective for reactive, shorter-term horizons, the

*Table 4.* Hyperparameters for Simulation (Adroit/Meta-World) and Real-world Experiments.

| Parameter | Simulation (Adroit/Meta-World) | Real-world |
|---|:---:|:---:|
| *Data & Architecture* | | |
| Demonstrations | 10 | 50 |
| Point Cloud Size (FPS) | 512 / 1024 | 1024 |
| Image Resolution | $84 \times 84$ | $84 \times 84$ |
| *Policy Config* | | |
| Observation Window ($T_{obs}$) | 2 | 2 |
| Prediction Horizon ($T_{pred}$) | **4** | **16** |
| Execution Horizon ($T_{exec}$) | **3** | **8** |
| *Optimization* | | |
| Training Epochs | 3000 (Adroit) / 1000 (MW) | 3000 |
| Batch Size | 128 | 128 |
| Optimizer | AdamW ($lr = 10^{-4}$) | AdamW ($lr = 10^{-4}$) |
| *Evaluation* | | |
| Random Seeds | 3 (0, 10, 20) | - |
| Eval Metric | Top-5 Avg Success | 20 Trials |

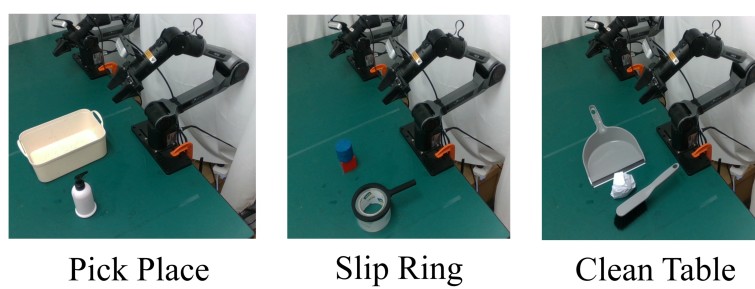

Pick Place       Slip Ring       Clean Table

*Figure 8.* **Tasks on Real Robots.** We evaluate our method on three physical robotic tasks: Pick and Place, where the robot must grasp a bottle and deposit it into a bin; Clean Table, which requires sweeping debris into a dustpan; and Slip Ring, where the robot slides a ring onto a cylinder.

DDE formulation may offer enhanced stability and consistency when the model is required to commit to longer sequences of future actions. Across all settings, the Hammer task remains saturated at 100% success, indicating that both variants are highly capable of mastering simpler manipulation trajectories regardless of the temporal horizon.

**Sensitivity across Different Action Chunks.** Regarding the performance of OMP-JVP across different temporal scales, the variant achieves its peak performance at the Short and Medium scales, both yielding an overall average success rate of 82.3%. At the Short scale, JVP demonstrates its highest efficacy in reactive tasks, particularly within the Meta-World Easy (89.7%) and Very Hard (77.8%) benchmarks. As the scale shifts to Medium, while the average remains stable, we observe a performance peak in more dexterous tasks such as the Pen task (66%) and Meta-World Hard tasks (67.3%), suggesting that a moderate increase in prediction horizon provides a beneficial temporal context for the Jacobian-based updates. However, performance notably declines at the Long scale ($H_p = 16, H_e = 8$), where the average success rate drops to 76.6%. The most significant degradation occurs in the Meta-World Easy tasks, which fall from 89.7% to 82.2%, indicating that the JVP formulation may be more susceptible to compounding errors or drifting when executing extended open-loop action sequences without frequent replanning.

In terms of temporal scaling, OMP-DDE exhibits a distinct performance profile compared to JVP. The variant begins with a strong baseline at the Short scale, achieving an 80.8% average success rate and showing particular proficiency in the high-dexterity Pen task (64%). A slight performance dip is observed at the Medium scale, where the average success rate decreases to 78.3%, primarily driven by declines in the Meta-World Easy and Medium benchmarks. Interestingly, OMP-DDE demonstrates a unique recovery at the Long scale, where it achieves its highest results in several complex

| Methods | Adroit | | | Meta-World | | | | **Average** |
|---|---|---|---|---|---|---|---|---|
| | Hammer | Door | Pen | Easy (21) | Medium (4) | Hard (4) | Very Hard (5) | |
| OMP-JVP(Short) | **100**±0 | **68**±3 | 60±4 | **89.7**±0.7 | **77.4**±2.2 | **62.5**±3.1 | **77.8**±3.0 | **82.3**±1.6 |
| OMP-DDE(Short) | **100**±0 | **68**±2 | **64**±3 | 89.0±1.3 | 76.4±2.7 | 61.0±3.0 | 70.6±4.9 | 80.8±2.2 |
| OMP-JVP(Medium) | **100**±0 | **64**±2 | **66**±3 | **86.3**±1.2 | **76.2**±2.5 | **67.3**±3.4 | **73.8**±3.1 | **80.7**±1.9 |
| OMP-DDE(Medium) | **100**±0 | 63±3 | 65±3 | 84.5±1.6 | 72.1±2.4 | 60.8±2.9 | 73.2±4.5 | 78.3±2.3 |
| OMP-JVP(Long) | **100**±0 | **69**±2 | 64±3 | 82.2±1.5 | **70.3**±2.7 | 59.5±3.0 | 71.3±3.8 | 76.6±2.1 |
| OMP-DDE(Long) | **100**±0 | 67±3 | **70**±3 | **83.4**±1.8 | 69.8±3.4 | **62.0**±2.7 | **75.2**±3.7 | **78.2**±2.4 |

*Table 5.* Performance comparison of JVP and DDE variants within Short, Medium, and Long action chunk settings.

categories, including the Pen task (70%) and Meta-World Hard tasks (62.0%). This resilience suggests that the DDE formulation is better suited for long-range trajectory planning, as it maintains superior consistency and stability when the execution horizon is extended to 8 steps, ultimately outperforming JVP in the Long setting.

### E.2. Sensitivity Analysis on $\epsilon$

| Methods | NFE | Adroit | | | Meta-World | | | | **Average** |
|---|---|---|---|---|---|---|---|---|---|
| | | Hammer | Door | Pen | Easy (21) | Medium (4) | Hard (4) | Very Hard (5) | |
| OMP-DDE ($\epsilon = 0.005$) | 1 | **100**±0 | **68**±2 | **64**±3 | 89.0±1.3 | **76.4**±2.7 | **61.0**±3.0 | **70.6**±4.9 | 80.8±2.2 |
| $\epsilon = 0.01$ | 1 | **100**±0 | 67±2 | 61±3 | **89.2**±1.4 | 74.2±3.1 | 58.5±2.7 | 68.8±4.2 | 80.4±2.1 |
| $\epsilon = 0.001$ | 1 | 94±3 | 57±3 | 58±4 | 87.2±1.9 | 74.5±2.6 | 57.8±4.2 | 68.4±4.5 | 78.7±2.7 |
| $\epsilon = 0.0005$ | 1 | 88±6 | 52±5 | 53±6 | 86.3±2.7 | 72.4±3.2 | 58.9±4.8 | 68.5±5.2 | 77.6±3.5 |
| $\epsilon = 0.0001$ | 1 | 80±5 | 52±4 | 50±4 | 84.2±2.2 | 70.9±3.0 | 56.4±4.1 | 66.1±4.2 | 75.3±2.9 |

*Table 6.* Sensitivity analysis of the hyperparameter $\epsilon$ on the Adroit and Meta-World benchmarks. The default configuration uses $\epsilon = 0.005$.

We evaluate the sensitivity of the OMP-DDE variant to the hyperparameter $\epsilon$ (Table 6). The default configuration of $\epsilon = 0.005$ yields the highest overall average success rate (80.8%) and optimal or near-optimal performance across all individual benchmark categories. Marginally increasing $\epsilon$ to 0.01 maintains competitive performance (80.4% average). In contrast, decreasing $\epsilon$ below 0.005 leads to a monotonic and significant degradation in success rates across all environments. This decline is particularly pronounced in dexterity-heavy tasks; for instance, performance on Adroit Hammer drops from 100% at $\epsilon = 0.005$ to 80% at $\epsilon = 0.0001$. These results demonstrate that while the system is relatively robust to minor increases in $\epsilon$, maintaining an adequate lower bound is critical for stable optimization and high-fidelity action generation.

### E.3. Failure cases in Real Experiments

The failure cases stem from two system constraints:

- Visual occlusions: As hypothesized, occlusion is a major factor. Our real-world setup relies on a single depth camera. During close-proximity manipulation, the robotic arm occasionally occludes the rings and target objects, which degrades the sampled point clouds and leads to imprecise spatial predictions.

- Open-loop action chunking: To maintain a high control frequency for efficient task execution, the policy predicts long, multi-frame action chunks. Because these chunks are executed strictly open-loop, the robot cannot dynamically adjust to small errors mid-trajectory. Consequently, minor residual misalignments or unexpected physical resistance upon contact cannot be corrected on the fly, leading to task failure.

