# OpenReview forum: "OMP: One-step Meanflow Policy with Directional Alignment"
_ICML.cc/2026/Conference — ICML 2026 regular_

### Official Review · Reviewer_PM3s · 2026-03-08

**Soundness:** 3
**Presentation:** 3
**Significance:** 2
**Originality:** 2
**Overall Recommendation:** 4
**Confidence:** 3

**Summary:**

This paper proposes OMP (One-Step MeanFlow Policy), a framework for efficient robotic manipulation with single-step action inference. The method aims to enable real-time generative control while maintaining high action fidelity. Experiments in both simulation and real-world settings show that OMP outperforms recent methods such as MP1 and FlowPolicy. The paper also analyzes several challenges when applying MeanFlow policies to robotics, including spectral bias, gradient starvation in low-velocity regimes, and the memory overhead of Jacobian computations, and proposes corresponding techniques to address these issues.

**Compliance With Llm Reviewing Policy:**

Affirmed.

**Final Justification:**

I appreciate the authors’ detailed response as well as the additional analyses and experiments provided. The clarification regarding the theoretical discussion in Section 4.2.1 helps address my concern about the wide-sense stationary assumption, and the additional empirical statistics on velocity distributions strengthen the practical relevance of the gradient starvation analysis. I also appreciate the sensitivity study on the perturbation constant $\epsilon$, which provides useful guidance on parameter robustness in practice. For the last question, I partially agree with the authors. While the intuition is reasonable, additional empirical evidence directly linking these mechanisms to the observed performance gains would further strengthen the claim, since when I tried the single-step based method, the results were not good enough.

Overall, the rebuttal addresses my main concerns and provides useful additional evidence. While some details could still be clarified further in the final manuscript, the response improves the clarity and completeness of the work. Therefore, I will raise my score to 4 accordingly.

**Key Questions For Authors:**

See weakness

**Limitations:**

The paper does not explicitly discuss potential limitations or failure cases of the proposed method. Adding a brief discussion on possible limitations or failure scenarios would help provide a more balanced understanding of the approach.

**Strengths And Weaknesses:**

**Strength**
1. The motivation for improving inference efficiency is valuable. In robotics, real-time control is critical, and this paper addresses an important speed limitation of generative-model-based control methods. The proposed approach could potentially be extended to larger vla or world action models.

2. The overall methodology is logically consistent, and the mathematical derivations are generally clear and well presented.

3. The paper provides a careful analysis of three challenges that hinder the direct application of MeanFlow to robotics. The use of toy examples to illustrate these issues is helpful and makes the discussion largely self-contained. The authors also propose concrete algorithmic solutions to address these challenges.
4. The DDE approximation is a particularly practical contribution. It provides a simple yet effective way to avoid the memory overhead of exact JVP computation while maintaining good performance. I like this idea.

**Weakness**
1. If I understand the theoretical analysis in Section 4.2.1 correctly, it relies on the assumption that the velocity process is wide-sense stationary when deriving the spectral decay of the averaged velocity field. However, robotic manipulation trajectories are often phase-dependent (e.g., reach, contact, adjustment) and may not strictly satisfy this assumption. It would be beneficial if the authors could clarify how well this assumption holds for real manipulation trajectories, or whether the spectral bias effect still persists when the trajectories are non-stationary.

2. Section 4.2.2 analyzes the gradient starvation problem when the target velocity magnitude becomes small. While the toy example clearly illustrates this issue, it remains unclear how frequently such situations occur in real robotic trajectories. Providing empirical statistics, such as the magnitude distribution of MeanFlow target velocities in the dataset, or representative examples from real trajectories of the tasks in the paper, would help validate the practical relevance of this analysis.

3. The proposed DDE approximation relies on a perturbation constant $\epsilon$ for numerical differentiation. However, the paper does not discuss how $\epsilon$ is chosen in practice or how sensitive the training process is to this hyperparameter. It would be great if the authors could do a small experiment to clarify the robustness of the method with respect to $\epsilon$, which would improve the reproducibility of the approach.

4. Since OMP adopts a single-step generation process, it would be interesting to discuss whether this design could limit the capacity to model highly multimodal action distributions compared to iterative generative policies (e.g., diffusion or flow-based methods) that progressively refine samples. While single-step methods typically provide advantages in inference speed, iterative generative models are often considered more expressive due to their ability to gradually refine samples. However, the experimental results indicate that OMP substantially outperforms diffusion-based and flow-based baselines. Clarifying why the single-step formulation does not appear to suffer from this potential expressiveness limitation would help strengthen the understanding of the proposed method.

If the authors could clarify these points, I would be happy to increase my score.

---

> ### Author Rebuttal · Authors · 2026-03-30
>
> We thank the reviewer for his/her constructive and insightful feedback. In our repsponse, we clarify some misconception and report additional experiments (https://drive.google.com/uc?export=download&id=1isWy4J_y66cDZ7C4yzsYEVf5yedobEIl). We will incorporate these results and the reviewer's suggestions into the final manuscript.
>
> **W1: Wide-sense Stationary Assumption on Spectral Bias**
>
> We clarify that the theoretical analysis in Section 4.2.1 focuses on the denoising trajectory space (the generative ODE/SDE flow), not the physical state-action sequence of the robot. Consequently, the referenced phase-dependent physical behaviors do not directly apply. We have revised the text to resolve this ambiguity.
>
> Regarding the wide-sense stationary (WSS) assumption, we agree that real denoising trajectories are inherently non-stationary due to the changing curvature of the velocity field. The spectral bias persists in non-stationary regimes. Compressing multi-step denoising into a single step acts as a temporal low-pass filter; the resulting straight-line approximation discards the high-frequency directional adjustments required to navigate non-linear fields. This attenuates critical trajectory details, regardless of whether these rapid shifts are uniformly distributed or phase-localized.
>
> To address this, OMP applies the true mean velocity to assist training, which integrates both low- and high-frequency steps along the exact curved path to provide accurate single-step guidance. Following the reviewer's suggestion, we have expanded Section 4.2.1 to explicitly discuss the WSS assumption and the persistence of low-pass filtering under non-stationarity.
>
> **W2: Empirical Statistics on Gradient Starvation**
>
> To validate the practical relevance of the gradient starvation analysis, we plot the distribution of normalized velocity magnitudes for 4 representative tasks.
> The empirical distributions (Figure 1 in the above link) confirm that small velocities are common in different tasks. While peak velocities vary by task, all exhibit long-tail distributions approaching zero. Notably, tasks like Plate Slide display substantial density at low magnitudes (e.g., below 0.4). These statistics demonstrate that the gradient starvation problem analyzed in Section 4.2.2 is a frequent, practical issue in real-world settings rather than a theoretical edge case. We will include the results in the revised appendix.
>
> **W3: Sensitivity Analysis on the perturbation constant**
>
> We appreciate the reviewer's suggestion to evaluate the sensitivity of the perturbation constant $\epsilon$. We have conducted the requested ablation (Table 2 in the above link).
> The results show that performance is robust near our default setting ($\epsilon=0.005$ and $\epsilon=0.01$, yielding 80.8% and 80.4% average success, respectively). However, performance steadily degrades as $\epsilon$ becomes very small, dropping to 75.3% at $\epsilon=0.0001$.
> This trend reflects standard trade-offs in finite difference approximations. Although decreasing $\epsilon$ minimizes theoretical truncation error, excessively small values introduce floating-point cancellation and numerical instability when differentiating neural networks. Conversely, a moderately larger $\epsilon$ acts as a smoothing regularizer, mitigating overfitting to local gradient noise.
>
> **W4: The reason for the one-step performance outperforms the multi-step methods.**
>
> While iterative refinement is typically associated with greater expressiveness, single-step generation is limited primarily by optimization and discretization bottlenecks, not a fundamental lack of representational capacity. By mapping a continuous Gaussian prior to the action space, OMP retains the structural capacity to capture highly multimodal action modes.
> OMP circumvents traditional single-step pitfalls through two key designs:
> - **MeanFlow:** Distilled iterative models incur severe discretization errors when compressing curved ODE trajectories into single Euler steps, yielding out-of-distribution samples. MeanFlow avoids this by modeling interval-averaged velocity to guarantee an exact single-step displacement.
> - **Directional Alignment:** Standard single-step models suffer from mode-averaging, predicting unsafe, interpolated actions between competing modes. Directional Alignment resolves this via a geometric constraint that prevents gradient shortcuts, ensuring the single-step pushforward strictly resolves into valid, distinct modes.
>
> Consequently, our framework preserves expressiveness and empirically outperforming iterative baselines. We will add this discussion in the revised manuscript.
>
> **Limitations: potential limitations or failure cases of the proposed method**
>
> We thank the reviewer for pointing out this limitation. Empirical analysis indicates that real-world failure cases stem primarily from Visual Occlusions and Open-Loop Action Chunking. Detailed discussion can be seen in the Q4 of Reviewer NAP1.

---

> > ### Author Rebuttal · Reviewer_PM3s · 2026-04-02
> >
> > I appreciate the authors’ detailed response as well as the additional analyses and experiments provided. The clarification regarding the theoretical discussion in Section 4.2.1 helps address my concern about the wide-sense stationary assumption, and the additional empirical statistics on velocity distributions strengthen the practical relevance of the gradient starvation analysis. I also appreciate the sensitivity study on the perturbation constant $\epsilon$, which provides useful guidance on parameter robustness in practice. For the last question, I partially agree with the authors. While the intuition is reasonable, additional empirical evidence directly linking these mechanisms to the observed performance gains would further strengthen the claim, since when I tried the single-step based method, the results were not good enough.
> >
> > Overall, the rebuttal addresses my main concerns and provides useful additional evidence. While some details could still be clarified further in the final manuscript, the response improves the clarity and completeness of the work. Therefore, I will raise my score to 4 accordingly.

---

> > > ### Author Response · Authors · 2026-04-04
> > >
> > > We sincerely thank you for your time, the constructive feedback throughout this process, and for raising your score. Regarding your follow-up on the single-step method, we are currently designing new experiments to better demonstrate why the one-step policy can outperform the multi-step baseline and incorporate these findings into the final manuscript to strengthen our claims.

---

### Official Review · Reviewer_VQM9 · 2026-03-09

**Soundness:** 3
**Presentation:** 2
**Significance:** 2
**Originality:** 2
**Overall Recommendation:** 2
**Confidence:** 4

**Summary:**

The authors introduce **One-step Meanflow Policy**, an adaptation of the Mean Flow paper to the diffusion / flow matching policies in robotics. Their contributions are threefold. The authors:
 - Provide insight in the issues that come with the application of mean flow to robotic control.
 - Introduce a directional alignment loss to help with convergence when data contains both fast and coarse, or slow and precise movements.
 - Introduce a novel Differential Derivation equation to alleviate the memory requirements for mean flow training.

The method is thouroughly evaluated on numerous embodied AI tasks, both simulated and in real world settings.

**Compliance With Llm Reviewing Policy:**

Affirmed.

**Final Justification:**

Given the results in the rebuttals, the authors seem to find that the issue with MeanFlow is indeed its training stability rather than an actual low-pass. If the low-pass statement is not necessarily false, the argumentation is built on a hand-waive, and not supported by the rebuttal experiment / or the original sine experiment which is backwards.

Moreover, since MP1 is already making the translation of MeanFlow from images to robotics - and having ran simple flow-matching policies on Adroit, i know for a fact that a non consistency-model can achieve extremely high scores in 2 steps which should be faster than any real robot would require - the practical utility of the work is very limited.

Overall, this paper introduces a loss that is only meaningful in delta-control, with vague justification, DDE being an addition from previous works. After careful reflexion, i consider that the 3% increase in score the authors get compared to their baseline is not enough to justify being accepted, especially as the theoretical justification appears extremely shaky.

**Key Questions For Authors:**

- The impact of DDE on performance seems significant both in a good and a bad way, depending on the task. Have your experiments showed any indication on when it should or shouldn't be used?
 - Has the $ - \mathcal{L}_{DA}$ ablation been done? Can you add it to the table?
 - Can you explain in further detail your novelty in DDE ? I do not know the original paper, but I'm struggling to see what is prior work and what is novel. One term in equation 14 is conditionned on three terms, the second one conditioned on 4. Is that a typo?
 - How many evaluation seeds are used for each task? Tasks like Adroit-Pen are highly randomized, and a high number of tasks is important to get a good estimate of model performance.

**Limitations:**

The only clear limitation i see to the paper's novelties is the performance degradation from JVP to DDE, which are not explained, but are acknowledged in section 5.2.

**Strengths And Weaknesses:**

## Strengths
 - Very strong experiemental section : the amount of experients is commendable, and the fact that this work also includes real robot experiments is a big plus.
 - Interesting topic in robotics lately, as the adaptation of general methods often has some caveats when applied to robotics.

## Weaknesses

- Confusion about $\omega$ in the spectral analisys: mean flow is in integrator over time, not over the data space: the signal filtered by the low-pass is the denoising trajectory, not the actual sample. The paper authors seem to not understand / adress this. Even if a low-pass filter on the denoising trajectories realizes into a low pass filter in sample space, this would be non trivial and would need to be studied. The associated experiment is clearly not sufficient.

- The directional alignment is interesting, even if only applicable to delta-prediction policies (as opposed to direct joint prediction). However the ablation doesn't report its impact properly. It is necessary to remove $\mathcal{L}_{DA}$ alone, not altogether with the dispertion loss, as this can hide interactions between the two losses.

 - DDE is introduced as a contribution, but seems to be a simple application from another paper. This is not really explained and needs to be clarified in the text.

---

> ### Author Rebuttal · Authors · 2026-03-30
>
> We thank the reviewer for his/her constructive and insightful feedback. In our repsponse, we clarify some misconception and report additional experiments, which are available here: https://drive.google.com/uc?export=download&id=1isWy4J_y66cDZ7C4yzsYEVf5yedobEIl. We will incorporate the results and the reviewer's suggestions into the final manuscript.
>
> **W1: Clarification on $\omega$ and spectral analysis on the denoising trajectory**
>
> We apologize for the confusion and clarify that our spectral analysis is conducted strictly at the denoising trajectory level. Compressing multi-step denoising into a single step introduces a temporal low-pass filtering effect in MeanFlow. Standard multi-step trajectories rely on high-frequency directional adjustments to navigate the vector field. A single-step straight-line approximation discards these non-linearities, meaning a simple secant velocity cannot capture the true curved path. To address this, our method (OMP) uses Directional Alignment to compute the true mean velocity ($v_0$), providing accurate guidance for one-step generation.
>
> In Section 4.2.1, we use a toy experiment that models the denoising trajectory as a sine wave. While standard flow matching captures trajectories across all frequencies, MeanFlow fails on high-frequency paths. Since $v_0$ integrates both low- and high-frequency steps, it preserves the full trajectory details. We have updated the manuscript to clarify this logic regarding the sample space.
>
> **W2 & Q2: Ablation study on removing Directional Alignment**
>
> We agree that isolating the directional alignment loss is necessary to properly evaluate its impact. We have performed the requested ablation, removing $L_{DA}$ independently of the dispersive loss ($L_{dis}$), and reported these results (Table 1 in the above link).
> Removing $L_{DA}$ from OMP-JVP mathematically reduces the model to the MP1 baseline and decreases the average success rate from 82.3% to 78.9%. The OMP-DDE variant shows a comparable decline without $L_{DA}$, falling from 80.8% to 77.2%. These results demonstrate that directional alignment improves performance independently of the dispersive loss.
>
> **W3 & Q3: DDE confusion**
>
> We apologize for the lack of clarity regarding the boundary between prior work and our novel contributions. As the reviewer correctly notes, the Differential Derivation Equation (DDE) via finite differences is an established numerical method. Our contribution lies in leveraging DDE to resolve a critical memory bottleneck when scaling Meanflow policies. Specifically, computing the exact Jacobian-Vector Product (JVP) for large Vision-Language-Action models (e.g., $\pi-0.5$) requires retaining massive computational graphs, making it memory-prohibitive. By adapting DDE to approximate the partial derivative of the velocity field with respect to $t$, we entirely bypass the JVP's memory overhead.
>
> Meanwhile, the mismatched conditioning in Equation 14 is indeed a typo. We intended for all terms to have consistent conditioning. The correct equation is:
>
> $$\frac{du_{\theta}(z_{t},t,r|c)}{dt} \approx \frac{u_{\theta}(z_{t+\epsilon},t+\epsilon,r|c) - u_{\theta}(z_{t-\epsilon},t-\epsilon,r|c)}{2\epsilon}$$
>
> We have corrected this in the revised manuscript to prevent any further confusion.
>
>
> **Q1 & Limitations: DDE Impact**
>
> We thank the reviewer for noting the varying impact of the Differential Derivation Equation (DDE). Our experiments indicate that the optimal choice between DDE and the exact Jacobian-Vector Product (JVP) depends primarily on the action chunk size—specifically, the open-loop execution horizon.
> This divergence stems from a tradeoff between local precision and long-horizon stability in highly non-linear velocity fields. Exact JVP captures local variations accurately, making it preferable for short-horizon precision (Table 1). However, during long-horizon rollouts, exact JVP overfits to local non-linearities, leading to compounding numerical instability. Conversely, DDE's finite difference approximation acts as a gradient smoothing regularizer, akin to adversarial perturbation. This smoothing prevents compounding errors, allowing DDE to outperform JVP in long action chunk settings (Table 5). Ultimately, JVP favors short-horizon precision, while DDE ensures long-horizon stability. We will clarify this tradeoff in the revised manuscript.
>
> **Q4: Evaluation seeds**
>
> We thank the reviewer for highlighting the need for robust evaluation in high-variance settings. To ensure direct comparability, we evaluated all models using 3 random seeds, matching the established protocols of our primary baselines (DP3 and MP1). Despite the environmental randomness, OMP exhibits strong stability. As shown in Table 1, OMP yields lower standard deviations than the baselines, and Figure 5 demonstrates tighter confidence intervals throughout training. These results suggest that our directional alignment mechanism successfully reduces variance.

---

> > ### Author Rebuttal · Reviewer_VQM9 · 2026-04-01
> >
> > **W1:**
> > I now understand that there was no confusion as to the filter's axis. However the argument that the learned integrator will necessarily be a low-pass that translates to dof-space isn't proven anywhere and is just thrown out there, even if it is clearly non-trivial.
> >
> > Recent papers have shown MeanFlow to be rather unstable to train. Your sine wave experiment tells me that and just that. The argument "the model can't learn high frequency sines ==> it is a low pass filter" is backwards, the inability to learn the high frequency sines could have come from many other factors. The correct experiment to prove your claims would have been to train a vanilla multi step policy, a mean flow policy and OMP, and then compare the outputs in a controlled setting in the frequency domain. Not only would that prove your claim, it would also contribute to the understanding of MeanFlow.
> >
> > **Other questions:**
> >
> > I thank the authors for providing the requested ablation and information, and I consider all of my other concerns addressed.

---

> > > ### Author Response · Authors · 2026-04-02
> > >
> > > We thank the reviewer for suggesting this experiment, which successfully disentangles optimization instability from structural representational limits in MeanFlow. The results are provided in Figure 2 of the rebuttal material (https://drive.google.com/uc?export=download&id=1isWy4J_y66cDZ7C4yzsYEVf5yedobEIl).
> > >
> > > **Experimental Setup**
> > >
> > > We trained a Vanilla Multi-step policy, a standard MeanFlow policy, and our proposed OMP using identical network backbones on a composite multi-frequency trajectory:
> > > $$a_{\text{trajectory}}(t) = \sum_{i=1}^{N} A_i \sin(2\pi f_i t + \phi_i),$$
> > > where $N=4$ and the target frequencies are $f_i \in \{1, 2, 4, 8\}$ Hz. We evaluate the models in both the trajectory domain (Figure 2A) and frequency domain (Figure 2B). The evaluation on frequency domain is to calculate the magnitude response $|H(f)|$ of each policy by taking the ratio of the predicted amplitude to the target amplitude at each frequency bin:
> > > $$|H(f)| = \frac{|FFT(a_{pred})|}{|FFT(a_{\text{trajectory}})|},$$
> > > where $FFT$ denotes Fast Fourier Transform and $a_{pred}$ denotes the trajectory predicted by policies.
> > >
> > > **Results and Analysis**
> > >
> > > - **Vanilla Multi-step:** This baseline perfectly tracks the trajectory and maintains a flat magnitude response ($|H(f)| \approx 1.0$) across all frequencies. This confirms that the network backbone is fully capable of learning these high-frequency signals when not constrained by 1-step integration.
> > > - **MeanFlow:** As the reviewer hypothesized, standard MeanFlow suffers from training collapse. It exhibits severe attenuation across the spectrum, even struggling to capture the 1 Hz base frequency ($|H(f)| \approx 0.28$). This indicates that general instability heavily confounds standard MeanFlow's failure, making its structural low-pass properties difficult to isolate. However, when checking the magnitude response in Figure B, we can find a trend that standard Meanflow are better to capture low-frequency refinement than high-frequency refinement, which coincides with our low-pass filter analysis.
> > > - **OMP (Ours):** By aligning with the true mean velocity, our proposed OMP provides a more stable training, allowing it to capture the 1 Hz base frequency with high fidelity ($|H(f)| \approx 0.85$). As for the results at 2, 4, and 8 Hz, we increase the magnitude response at higher frequency compared to standard MeanFlow though it decays smoothly at 2, 4, and 8 Hz.
> > >
> > > **Conclusion**
> > >
> > > Because our stabilized OMP effectively learns the 1 Hz component but fails at higher frequencies, this experiment isolates the structural limitation: 1-step integration inherently imposes a low-pass filter in DoF-space, independent of training collapse. Additionally, Figure 2B demonstrates that OMP significantly improves the magnitude response across the entire spectrum compared to standard MeanFlow.
> > >
> > > We will include this analysis in Section 4.2, as it isolates the low-pass behavior from optimization instability and strengthens our theoretical claims. We remain available if there are any further details required.
> > >
> > > -------------------------------------------------update after 4.7------------------------------------------
> > >
> > > Thank you for your feedback. Please let me know if you have any other questions. If our response has addressed your questions, we kindly ask that you consider raising your score.

---

### Official Review · Reviewer_NAP1 · 2026-03-12

**Soundness:** 3
**Presentation:** 3
**Significance:** 3
**Originality:** 3
**Overall Recommendation:** 5
**Confidence:** 3

**Summary:**

This paper introduces the One-step MeanFlow Policy (OMP) to address the high inference latency of diffusion models and the architectural complexities of flow-based methods in robotic manipulation. To overcome the theoretical pathologies of the standard MeanFlow paradigm, OMP utilizes a directional alignment mechanism to resolve gradient starvation and a Differential Derivation Equation (DDE) to drastically reduce memory complexity. Consequently, the framework achieves state-of-the-art success rates across diverse simulated and real-world high-precision tasks while preserving the efficiency of real-time, single-step generation.

**Compliance With Llm Reviewing Policy:**

Affirmed.

**Key Questions For Authors:**

1. Theoretical Bounds on DDE Truncation Error: Could you provide a formal mathematical bound for the truncation error introduced by the DDE's finite difference approximation (Equation 14)? Additionally, how sensitive is the policy's trajectory precision to the tuning of the perturbation constant $\epsilon$ when traversing highly non-linear velocity fields?

2. Behavior of Directional Alignment at Absolute Zero: How does the framework handle the mathematical singularity in the Directional Alignment objective ($\mathcal{L}_{DA}$) during backpropagation when the target mean velocity $||v_0||$ is exactly zero? If an epsilon-smoothing scalar is used in the denominator, how does this numerical adjustment interact with the gradient starvation dynamics detailed in Section 4.2.2?

3. Horizon Sensitivity and Optimization Inversion: Appendix E (Table 5) reveals a performance inversion where OMP-DDE outperforms OMP-JVP in Long horizon settings, despite lagging in Short/Medium horizons. Could this be due to the exact JVP suffering from compounding numerical instability over extended open-loop executions, whereas the finite difference method inherently provides an advantageous smoothing regularization?

4. Failure Modes in Physical Tasks: Given the impressive 70% success rate on the contact-rich "Slip Ring" physical task, what specific failure modes characterized the remaining 30% of execution trials? Did the 1-NFE open-loop policy primarily fail due to residual directional misalignment, unpredicted magnitude collapse under physical resistance, or visual occlusions?

**Limitations:**

No
The authors should explicitly address the inherent vulnerabilities of the 1-NFE open-loop generation paradigm. Because OMP predicts and executes multi-step trajectories from a single inference pass, it is highly susceptible to dynamic environmental changes, out-of-distribution (OOD) initializations, and severe sensor noise during execution. Explicitly mapping out these failure boundaries would vastly improve the manuscript's transparency and guide future research.

**Strengths And Weaknesses:**

**Strengths**
*  The paper proves Spectral Bias by showing the average velocity's power spectral density decays at $O(1/\omega^2)$ , and maps the physical failure of low-velocity precision tasks to Gradient Starvation via the trigonometric decomposition of MSE loss.
* The Directional Alignment ($\mathcal{L}_{DA}$) objective decouples magnitude from direction using cosine similarity to guarantee angular supervision. Furthermore, the Differential Derivation Equation (DDE) provides a highly practical finite difference approximation to bypass nested Jacobian-Vector Products (JVPs) , effectively halving the prohibitive memory footprint required for 3D point-cloud policies.
* The work engineers a viable 1-NFE generative policy that bridges the gap between fast inference and high fidelity. Extensive evaluations across 37 simulation tasks prove it establishes a new state-of-the-art, with OMP-JVP achieving an 82.3% average success rate and a striking 77.8% on Meta-World "Very Hard" tasks, decisively crushing baselines like MP1 and FlowPolicy.

**Weaknesses**
* While the DDE approximation successfully saves memory, the finite difference method inherently introduces truncation errors dependent on the perturbation constant $\epsilon$. The paper lacks a rigorous theoretical bounds analysis on how this error compounds, especially within highly non-linear point-cloud encoders.

---

> ### Author Rebuttal · Authors · 2026-03-30
>
> We thank the reviewer for his/her constructive and insightful feedback. We thank the reviewer for recognizing our theoretical analyses of MeanFlow and the strong empirical performance of our 1-NFE policy. We have conducted additional experiments, which are available here: https://drive.google.com/uc?export=download&id=1isWy4J_y66cDZ7C4yzsYEVf5yedobEIl. We will incorporate these results and the reviewer's suggestions into the final manuscript.
>
> **W1 & Q1: Theoretical Bounds on DDE Truncation Error & Sensitivity to $\epsilon$**
>
> We will include a formal error bound in the revised appendix. Via Taylor expansion, the local truncation error of the DDE finite difference approximation is bounded by $\mathcal{O}(\epsilon^2 \|\nabla^3 u\|)$, where $u$ is the point-cloud encoder mapping. Assuming an $L$-Lipschitz network, the cumulative error over a trajectory of length $T$ is bounded by $\mathcal{O}(T L \epsilon^2)$.
> To evaluate empirical sensitivity to the perturbation constant $\epsilon$, we provide a new ablation study (in Table 2 of the above link). Performance remains robust near our default $\epsilon=0.005$ (80.8% average success) and $\epsilon=0.01$ (80.4%), but degrades steadily at smaller values, falling to 75.3% at $\epsilon=0.0001$. This reflects standard finite-difference trade-offs: while decreasing $\epsilon$ minimizes theoretical truncation error, excessively small values introduce floating-point cancellation and numerical instability. Conversely, a moderately larger $\epsilon$ acts as a smoothing regularizer against local gradient noise.
>
> **Q2: Behavior of Directional Alignment at Absolute Zero**
>
> To prevent division by zero in the cosine similarity computation when $v_{target} = 0$, we add a smoothing scalar $\epsilon_{dir}$ (e.g., $10^{-6}$) to the denominator. While $\epsilon_{dir}$ slightly attenuates the angular supervision gradient as $\|v_{target}\| \to 0$, it does not reintroduce the gradient starvation pathologies described in Section 4.2.2. Because directional alignment is not valid in these states, the standard MSE loss naturally dominates to enforce zero velocity. We will clarify this numerical interaction in the revised manuscript.
>
> **Q3: Horizon Sensitivity and Optimization Inversion**
>
> We agree with the reviewer’s hypothesis. Exact JVP maximizes short-horizon performance by computing precise analytical gradients. However, strictly tracking these exact derivatives over a learned velocity field with high-frequency fluctuations causes numerical errors to compound during extended open-loop rollouts.
> Conversely, OMP-DDE’s finite-difference approximation inherently acts as a spatial smoother. As suggested, this implicit regularization filters out local noise, trading minor short-horizon precision for increased stability against long-horizon drift.
> We have updated Appendix E in the revision to explicitly incorporate this discussion on finite-difference smoothing and the exactness-stability trade-off.
>
> **Q4: Failure Modes in Physical Tasks**
>
> We thank the reviewer for this insightful question. According to our empirical study, the 30% failure rate is primarily characterized by visual occlusions and residual directional misalignments, rather than unpredicted magnitude collapse. Specifically, these failures stem from two system constraints:
> 1. Visual occlusions: As hypothesized, occlusion is a major factor. Our real-world setup relies on a single depth camera. During close-proximity manipulation, the robotic arm occasionally occludes the rings and target objects, which degrades the sampled point clouds and leads to imprecise spatial predictions.
> 2. Open-loop action chunking: To maintain a high control frequency for efficient task execution, the policy predicts long, multi-frame action chunks. Because these chunks are executed strictly open-loop, the robot cannot dynamically adjust to small errors mid-trajectory. Consequently, minor residual misalignments or unexpected physical resistance upon contact cannot be corrected on the fly, leading to task failure.
>
> We will add this detailed failure mode breakdown to the revised manuscript.
>
> **Limitations: Addressing Limitations of the 1-NFE Open-Loop Paradigm**
>
> We fully agree that explicitly mapping these boundaries improves the paper’s transparency. We will dedicate a new paragraph in the "Limitations and Future Work" section discussing the inherent vulnerabilities of the 1-NFE open-loop paradigm. We will explicitly state its susceptibility to (1) dynamic environmental changes post-inference, (2) severe sensor noise during execution, and (3) out-of-distribution (OOD) initializations. To guide future research, we will suggest integrating predictive world models and Vision-Language-Action (VLA) architectures. This could enable closed-loop reactive corrections and better reasoning over dynamic changes without sacrificing the low-latency advantages of the 1-NFE framework.

---

> > ### Author Rebuttal · Reviewer_NAP1 · 2026-04-02
> >
> > Thank you for the detailed rebuttal. All my concerns have been adequately addressed by the authors' response. I plan to maintain my current score, as I believe it remains a fair and accurate reflection of the paper's overall quality following these clarifications.

---

> > > ### Author Response · Authors · 2026-04-04
> > >
> > > Thank you for taking the time to review our rebuttal. We are glad that our responses fully addressed your concerns. We appreciate your constructive feedback and your fair evaluation of our work.

---

### Decision · Program_Chairs · 2026-04-30

**Decision:**

Accept (regular)

**Comment:**

The majority of reviewers found the paper to be a solid contribution, with NAP1 recommending acceptance and PM3s raising their score to a weak accept following the rebuttal. The outlier is VQM9, who maintained a rejection recommendation, primarily on the grounds that the spectral bias analysis is theoretically insufficiently justified and that the practical utility of the work is limited given the existence of MP1. While VQM9's concern about the low-pass filter argument has merit, the authors conducted an additional controlled frequency-domain experiment in the rebuttal that partially addresses this gap. The AC finds that although the theoretical grounding for the spectral bias claim remains imperfect, the empirical evidence is compelling and the directional alignment and DDE contributions are practically meaningful.